# Differential lateral and basal tension drive folding of *Drosophila* wing discs through two distinct mechanisms

Liyuan Sui[1], Silvanus Alt[2,3,8], Martin Weigert[4,5], Natalie Dye[5], Suzanne Eaton[5,6], Florian Jug [4,5], Eugene W. Myers[4,5,7], Frank Jülicher[2,4], Guillaume Salbreux[2,3] & Christian Dahmann[1]

Epithelial folding transforms simple sheets of cells into complex three-dimensional tissues and organs during animal development. Epithelial folding has mainly been attributed to mechanical forces generated by an apically localized actomyosin network, however, contributions of forces generated at basal and lateral cell surfaces remain largely unknown. Here we show that a local decrease of basal tension and an increased lateral tension, but not apical constriction, drive the formation of two neighboring folds in developing *Drosophila* wing imaginal discs. Spatially defined reduction of extracellular matrix density results in local decrease of basal tension in the first fold; fluctuations in F-actin lead to increased lateral tension in the second fold. Simulations using a 3D vertex model show that the two distinct mechanisms can drive epithelial folding. Our combination of lateral and basal tension measurements with a mechanical tissue model reveals how simple modulations of surface and edge tension drive complex three-dimensional morphological changes.

[1] Institute of Genetics, Technische Universität Dresden, 01062 Dresden, Germany. [2] Max Planck Institute for the Physics of Complex Systems, Nöthnitzer Strasse 38, 01187 Dresden, Germany. [3] The Francis Crick Institute, 1 Midland Road, NW1 1AT London, UK. [4] Center for Systems Biology Dresden (CSBD), Pfotenhauerstrasse 108, 01307 Dresden, Germany. [5] Max Planck Institute of Molecular Cell Biology and Genetics, Pfotenhauerstrasse 108, 01307 Dresden, Germany. [6] Biotechnologisches Zentrum, Technische Universität Dresden, Tatzberg 47/49, 01309 Dresden, Germany. [7] Department of Computer Science, Technische Universität Dresden, 01062 Dresden, Germany. [8] Present address: Max-Delbrück-Center for Molecular Medicine, Robert-Rössle-Strasse 10, 13125 Berlin, Germany. These authors contributed equally: Liyuan Sui, Silvanus Alt. Correspondence and requests for materials should be addressed to G.S. (email: guillaume.salbreux@crick.ac.uk) or to C.D. (email: christian.dahmann@tu-dresden.de)

Epithelial sheets adopt complex three-dimensional shapes through a sequence of folding steps during animal development[1–3]. Epithelial folding is instrumental during processes such as embryonic gastrulation[4] and neural tube[5] and eye[6] formation, and defects in epithelial folding can lead to severe developmental disorders in humans[7].

Epithelial folding relies on the generation of mechanical forces that leads to coordinated cell shape changes[8]. Epithelial folding has been commonly attributed to apical constriction that is mediated by pulsatile contractions of an actomyosin network located beneath the cell apex[1,2,9–11]. Additional mechanisms such as cell rounding during mitosis[12], force generation by apoptotic cells[13], basolateral contractility[14], microtubule network remodeling[15], and modulation of the basal extracellular matrix (ECM)[16] contribute to epithelial folding. However, mechanical forces exerted at basal or lateral cell edges have not been measured and, thus, their contributions to epithelial folding remained unclear.

The larval *Drosophila* wing imaginal disc, an epithelium that gives rise to the future notum, hinge, and wing blade of adult flies, is an excellent model system to study morphogenesis[17]. The prospective hinge region of the wing imaginal disc forms three stereotypic folds:[18] a fold between the prospective notum and hinge regions, a central hinge fold (herein referred to as H/H fold), and a fold between the prospective hinge and pouch (which gives rise to the wing blade; H/P fold; Fig. 1a, Supplementary Figure. 1a-l). The mechanisms that position these folds have been studied[19–22], however, the mechanical forces that drive formation of these folds are unknown.

In this work, we focus on the underlying mechanical processes leading to the H/H and H/P folds. We show that the formation of the H/H fold involves a local decrease of ECM density resulting in decreased basal edge tension and the basal widening of cells. The formation of the H/P fold is characterized by fluctuations of F-actin at the lateral cell surface that are associated with increased lateral surface tension and a decrease in cell height. Our work uncovers contributions of basal and lateral tensions to epithelial folding.

## Results

**Cells widen basally during hinge fold formation**. To analyze the overall three-dimensional shape changes during H/H and H/P fold formation over time, we developed a protocol for live imaging of wing imaginal discs in culture (Methods). Cultured wing imaginal discs sustained cell proliferation for at least 10 h (Supplementary Fig. 1m, Supplementary Movie 1) and formed H/H and H/P folds with no visible difference in shape from the hinge folds of fixed wing imaginal discs (Supplementary Figure 1n–q, Supplementary Movie 1). Regions involved in the formation of the future folds were imaged in early-third instar wing imaginal discs (72 h after egg lay (AEL)) expressing Indy-GFP[23] to visualize cell membranes (Fig. 1b–g).

To analyze cellular shapes during the formation of the H/H and H/P folds, we generated red fluorescent protein (RFP)-marked clones of cells in wing imaginal discs expressing Indy-GFP and subsequently imaged the wing imaginal disc in culture (Supplementary Figure 2a–d, Supplementary Movie 2). The apical and basal outlines of single RFP-marked cells located at the center of folds were then manually tracked over time in cross sections perpendicular to the fold direction (Methods). The apical and basal tissue outlines were identified based on Indy-GFP (Supplementary Figure 2a–d). Cell shape and tissue morphology were characterized by a set of geometric parameters (Fig. 1h, i). During the first 200 min of folding, the H/H and the H/P folds underwent pronounced apical indentations at similar velocities (Fig. 1j–o). The indentations of the basal tissue surfaces were in

opposite direction between the two folds (Fig. 1n, o). The average basal cross-sectional length (basal length) was increasing in both folds, but this increase was more pronounced in the H/H fold (Fig. 1p, q). Consistently, basal cross-sectional area of cells in H/H folds notably increased over time (Supplementary Figure 2m), indicating that in particular the H/H fold cells widen at their basal side. Surprisingly, the average apical cross-sectional length (apical length) of cells stayed almost constant in both folds (Fig. 1p, q). Moreover, the apical cross-sectional area (apical area) of cells located in the center of the emerging folds remained roughly constant (Supplementary Fig. 2e–l). Similarly, cell volume (Methods) approximately remained constant (Supplementary Fig. 2n–t). We conclude that formation of the H/H and the H/P folds takes place in the absence of cell volume change and it does not occur through apical constriction, but rather involves widening of the basal side of cells.

**Cell proliferation is not required for fold formation**. Differences in the rate of cell proliferation may lead to tissue compression resulting in folding[24]. Moreover, cell rounding during mitosis can accelerate epithelial invagination[12]. To test whether differences in cell proliferation rate or cell proliferation itself were required for H/H or H/P fold formation, we analyzed cell proliferation rates in the notum and the pouch region of the wing imaginal disc at 68 h AEL. Cell proliferation rates were not significantly different between the two regions (Fig. 2a, b). Moreover, temporarily blocking cell division by using a temperature-sensitive allele of the Cyclin-dependent-kinase $Cdk1$[25] ($Cdk1^{E1-24}$) resulted in a timely fold formation (Fig. 2c–j, Supplementary Movie 3), showing that cell proliferation is not required for the formation of the H/H and H/P folds.

**Basal tension is higher than apical tension outside folds**. Since folding is not triggered by apical constriction or compression arising from cell division, we tested whether forces generated in cells below the apical plane contribute to the mechanics of folding. We observed throughout the wing imaginal disc an enrichment of F-actin and non-muscle Myosin II along basal cell edges, similar to the previously described actomyosin-rich apical epithelial belt (Fig. 3a–h)[26]. To test whether line tensions are generated in this basal network, we ablated single basal cell edges visualized by Indy-GFP with a focused laser beam before and during the time of folding and quantified the resulting recoil (68–76 h AEL; Fig. 3i–l, Methods). For comparison, we ablated cell edges at the level of adherens junctions. As a relative measure of mechanical tension, we measured the average recoil velocity within 0.25 s after ablation[27] (see Supplementary Methods). The average recoil velocity of ablated basal cell edges was about 3–5 times higher than the average recoil velocity of ablated apical cell edges (Fig. 3m, Supplementary Figure 3, Supplementary Figure 4, Supplementary Movie 4). Average recoil velocities were decreased following application of drugs inhibiting actin polymerization and myosin activity, both apically and basally (Supplementary Figure 3, Supplementary Figure 4, Supplementary Figure 5). These data indicate that basal edge tension is significantly higher than apical edge tension in the wing imaginal disc pouch outside the folds.

**Basal tension depends on ECM**. Because of the apparently similar structure of the apical and basal F-actin cortex (Fig. 3a–h), we wondered how the basal tension is increased as compared to the apical tension. The ECM can contribute to cell and tissue shape in epithelia[28]. To test whether the ECM influences basal edge tension, we treated 76 h AEL wing imaginal discs with collagenase to deplete the collagen network. Collagen was rapidly

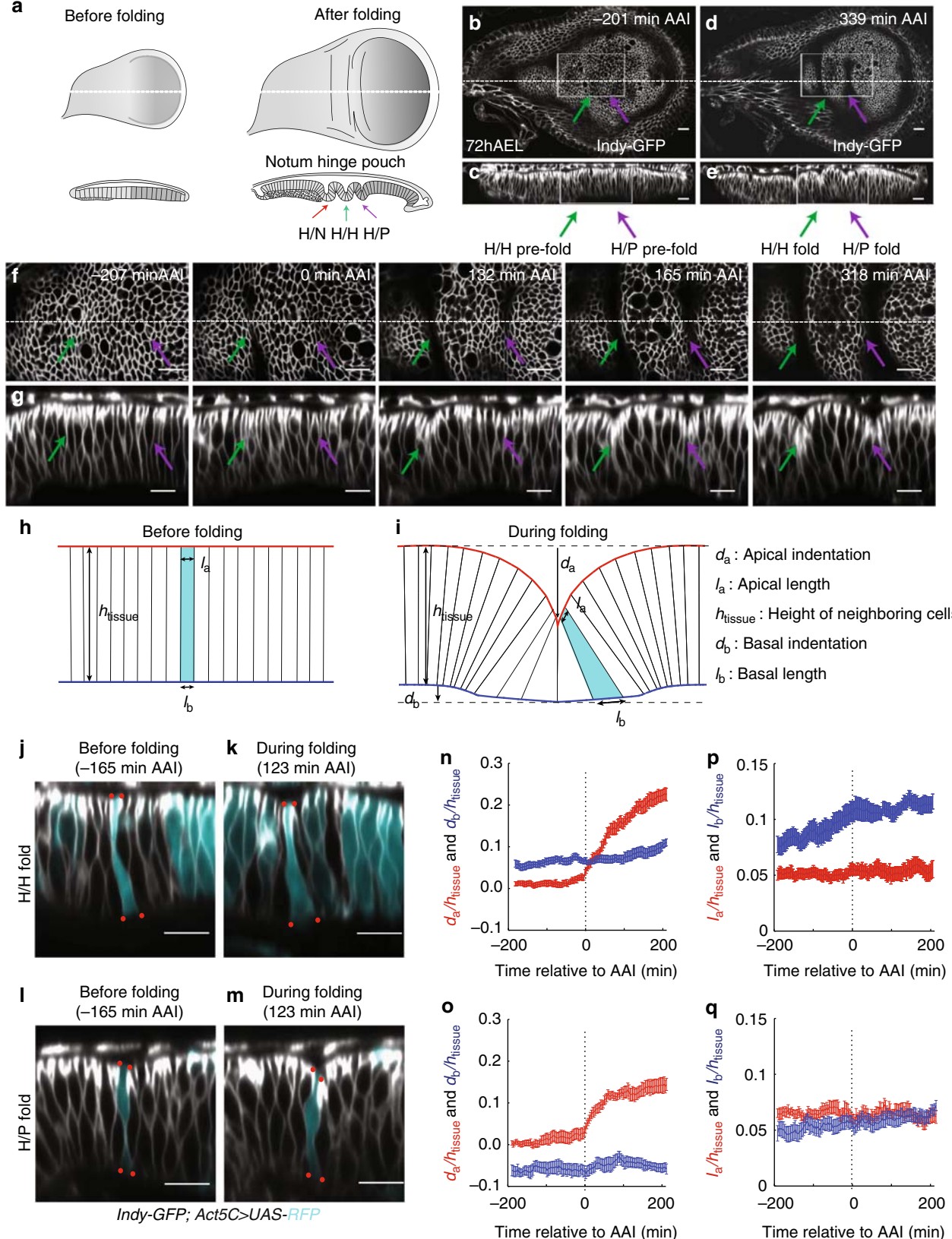

removed, as visualized using Viking-GFP, a green fluorescent protein (GFP) trap in the Collagen IV α2 chain[29] (Fig. 4a–h, Supplementary Movie 5). Both H/H and H/P folds were lost (Fig. 4a–h, Supplementary Movie 5). Moreover, wing imaginal disc cells increased their basal area, while the apical area did not

change as strongly (Fig. 4d, h, i). This observation suggested to us that the ECM has an impact on the basally generated tensions. To test this hypothesis further, we ablated apical and basal cell edges of wing imaginal discs at 72 h AEL before and after collagenase treatment and measured the resulting recoil. The average recoil

**Fig. 1** Quantitative analysis of cell shape changes during fold formation. **a** Schemes representing top views (above) and cross-sectional views (below) of wing imaginal discs before and after folding. The type of fold is indicated. **b–e** Top view (**b, d**) and cross-sectional (**c, e**) images of a time-lapse movie of a cultured 72 h AEL wing imaginal disc expressing Indy-GFP, showing formation of hinge-hinge (H/H) and hinge-pouch (H/P) folds. Time relative to first appearance of apical indentation (AAI) (i.e. the first time when the apical surface of fold cells is below the apical plane of neighboring cells) of H/H fold is shown. In this and the following figures, top views are shown with dorsal to the left and posterior up; in cross sections, the apical surface of columnar cells is to the top, unless otherwise indicated. Dotted lines in top views indicate position of the corresponding cross sections. Scale bars are 10 μm. **f, g** Top view (**f**) and cross-sectional (**g**) images of the boxed areas of the time-lapse movie shown in **b** and **d** at indicated time points. Scale bars are 10 μm. **h, i** Schemes showing simplified cell shapes before and during folding and the set of geometric parameters used. $d_a$ and $d_b$ denote the apical and basal deformations, $l_a$ and $l_b$ denote the apical and basal cross-sectional lengths of cells located at the center of the fold, and $h_{tissue}$ denotes the apico-basal height of cells neighboring the fold. **j–m** Cross-sectional images of a time-lapse movie of a cultured wing imaginal disc expressing Indy-GFP (gray) in all cells and RFP (turquoise) in clones of cells of H/H fold (**j, k**) or of H/P fold (**l, m**). Red dots mark apical and basal vertices of RFP-labeled cells. Scale bars are 10 μm. **n–q** Changes of the geometric parameters indicated in **i** during H/H (**n, p**) and H/P (**o, q**) fold formation as a function of time relative to AAI. All geometrical quantities are normalized by the cell height $h_{tissue}$ of the surrounding tissue. Mean and s.e.m. are shown. $n = 17$ cross sections of 7 wing imaginal discs for **n** and **p** and $n = 12$ cross sections of 6 wing imaginal discs for **o** and **q**

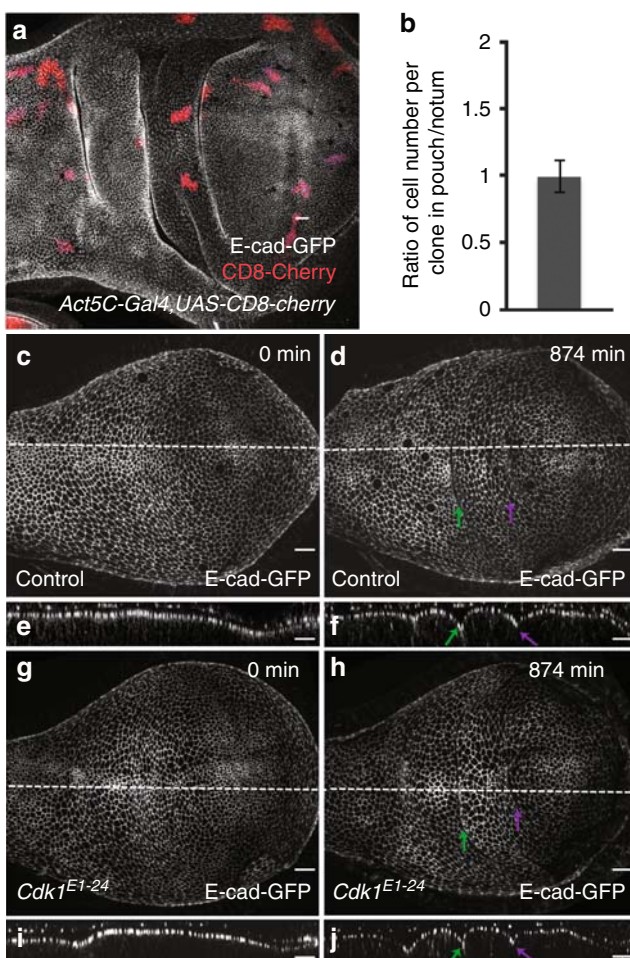

**Fig. 2** Cell proliferation and the role of cell division for epithelial folding. **a** A wing imaginal disc of a 96 h after egg lay (AEL) larva carrying 48 h-old clones of cells marked by the expression of CD8-mCherry (*Act5C > Gal4, UAS-CD8-mCherry*, red). Adherens junctions are labeled by E-cad-GFP (gray). Scale bar is 10 μm. **b** Ratio of the average cell number per clone in the pouch and the average cell number per clone in the notum. Mean and s. e.m. are shown. $n = 19$ wing imaginal discs, 82 clones in the pouch region, and 59 clones in the notum region. **c–j** Top view (**c, d, g, h**) and cross-sectional (**e, f, i, j**) images of time-lapse movies of control (**c–f**) and $Cdk1^{E1-24}$ mutant (**g–j**) cultured wing imaginal discs expressing E-cad-GFP are shown for the indicated time points after shift to the restrictive temperature. Scale bars are 10 μm

velocity upon ablating apical cell edges was not affected by collagenase treatment (Fig. 4j). By contrast, collagenase treatment reduced the average recoil velocity following the ablation of basal cell edges approximately threefold (Fig. 4j). We conclude that basal edge tension depends on ECM.

**Decreased collagen IV and basal tension in H/H fold.** To test whether basal edge tension plays a role in the formation of the H/H fold, we ablated single edges of H/H pre-fold cells. While the average recoil velocity after ablation of apical cell edges did not significantly change in H/H cells before or during folding, the average recoil velocity upon ablation of basal cell edges was reduced by about 70% at 72–76 h AEL, compared to recoil velocities measured in the pouch (Fig. 5a, Supplementary Figure 3, Supplementary Figure 4).

We then asked how the decrease in basal edge tension is triggered in the H/H fold. Since basal edge tension depends on ECM, we visualized ECM in the H/H fold region using Viking-GFP. Viking-GFP intensities were homogeneous underneath the epithelium outside the H/H fold but were reduced by approximately 20% underneath the H/H fold cells in a stripe of approximately four cells wide compared to neighboring cells (Fig. 5b–h). Integrin levels were also reduced in pre-fold H/H cells (Supplementary Figure 6a-i). Taken together, these findings suggest that the reduction of ECM in H/H pre-fold cells triggers the local decrease of basal edge tension in these cells.

**Local ECM reduction drives ectopic tissue folding.** To test whether the local reduction in ECM levels and the resulting reduction in basal edge tension are sufficient for epithelial folding in wing imaginal discs, we expressed matrix metalloproteinase II (MMP2), an extracellular protease that cleaves ECM components[30], in a stripe of cells along the anteroposterior compartment boundary. Integrin levels were reduced at the basal side of wing imaginal disc cells expressing MMP2 (Supplementary Figure 6j,k). Basal recoil velocity, but not apical recoil velocity, was significantly reduced before folding in MMP2-expressing cells (Fig. 5i). The reduction of basal recoil velocity in MMP2-expressing cells had a similar magnitude to the reduction observed following collagenase treatment (compare Figs. 5i, 4j), suggesting that it resulted from ECM degradation. Strikingly, cells expressing MMP2 became part of an epithelial fold that was absent in control wing imaginal discs (Fig. 5j)[20,31]. These results demonstrate that a local reduction of ECM components is sufficient for epithelial folding. Taken together, we conclude that during H/H fold formation the local reduction of ECM triggers a local decrease of basal edge tension driving the relaxation of the basal cell edges and tissue folding.

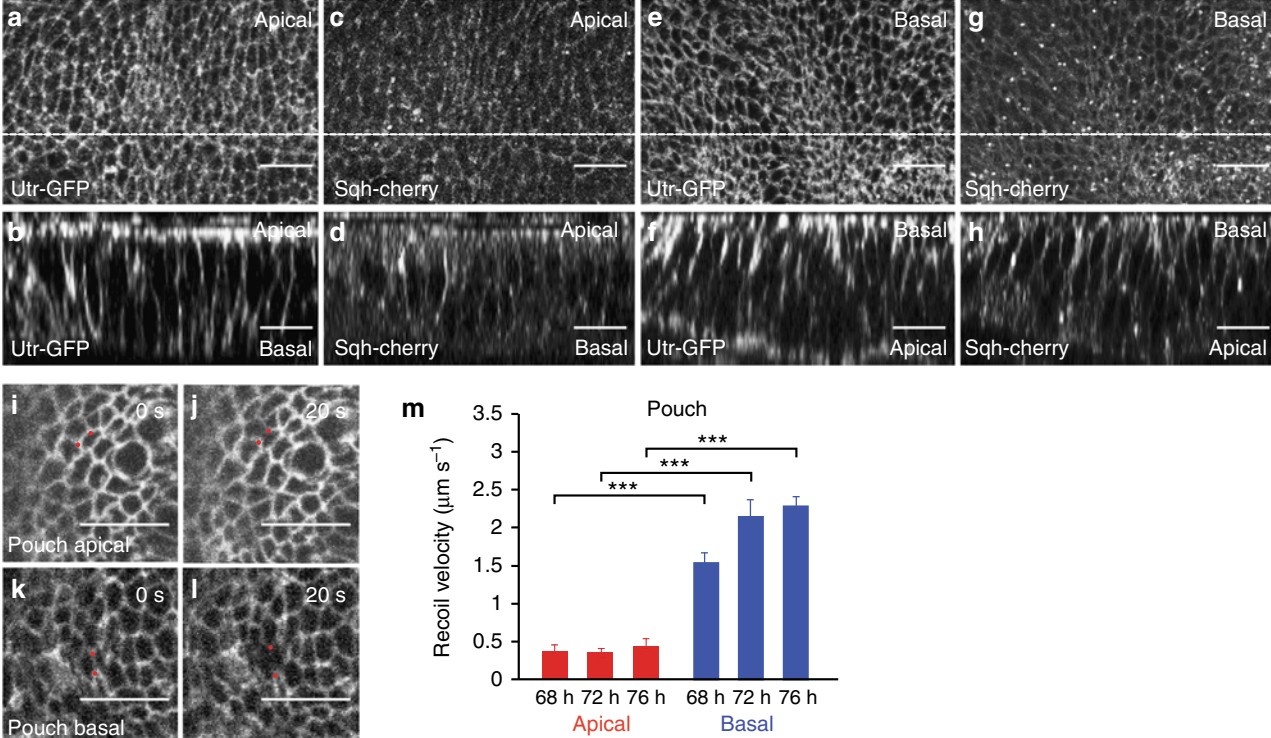

**Fig. 3** Basal tension is higher than apical tension outside folds. **a–h** Apical (**a**, **c**) and basal (**e**, **g**) views and cross-sectional images (**b**, **d**, **f**, **h**) of wing imaginal discs of 72 h AEL larvae co-expressing Utr-GFP and Sqh-cherry to visualize F-actin and Myosin regulatory light chain, respectively. The apical and basal sides of the columnar cells are indicated in the cross sections. In **a–d** the apical side of the columnar cells was mounted closer to the coverslip, whereas in **e–h** the basal side was mounted closer to the coverslip. Scale bars are 10 μm. **i–l** Wing imaginal disc pouch cells of 72 h AEL larvae expressing Indy-GFP before and 20 s after ablation of a single cell edge at the apical (**i**, **j**) or basal (**k**, **l**) side of the pouch epithelium. Scale bars are 10 μm. Red dots mark vertices of ablated cell edges. **m** Average recoil velocity of the two vertices at the end of an ablated cell edge within 0.25 s after ablation in the pouch region for wing imaginal discs of the indicated times AEL. Recoil velocities are shown for ablations of apical and basal cell edges, as indicated. Mean and s.e. m. are shown ($n = 15$ cuts) (***$p < 0.001$, Student's $t$-test)

**Increased F-actin and lateral tension in H/P fold**. While the H/P fold forms only shortly after the H/H fold (Fig. 1b–g), we did not observe a local reduction in collagen underneath pre-fold H/P cells (Fig. 5b–g), nor a reduction of recoil velocity upon ablation of basal cell edges in the H/P fold (Supplementary Figure 7a, Supplementary Figure 3), indicating that the H/P fold and H/H fold form by different mechanisms. We noted, however, that Utr-GFP imaging revealed a highly dynamic accumulation and flow of F-actin along the lateral interfaces of H/P fold cells and pulsatile contractions of their apical-basal height (Fig. 6a–d, Supplementary Movie 6). This was not the case for cells in the H/H fold (Supplementary Figure 7b–d).

To test if lateral F-actin accumulation is driving H/P fold cell deformations, we first quantified apical-basal cell height and average lateral F-actin intensity in cross sections (Fig. 6d, e, Methods). We then quantified the cross correlation between changes in F-actin intensity and cell height, and found a negative peak for a time lag around 22 s, indicating that an increase in lateral F-actin is closely followed by a decrease in cell height (Fig. 6f). Height and F-actin fluctuations in H/H fold cells were much weaker and did not exhibit a similar cross correlation (Supplementary Figure 7d–e). On timescales longer than the characteristic time of the pulsatile F-actin increase (minutes), the H/P fold cell height was decreasing, suggesting that this decrease contributes to H/P fold formation (Fig. 6d). H/P fold cells also displayed a highly dynamic accumulation of F-actin at their apical and basal areas that cross correlated with apical and basal cell constriction, respectively (Supplementary Figure 7f–l, Supplementary Movies 7 and 8). These constrictions, however, were not

as strongly correlated with changes in cell height (Supplementary Figure 7m–p). To test whether the accumulation of F-actin at lateral cell interfaces correlates with increased lateral surface tension, we developed a method to perform laser ablation experiments cutting lateral cell interfaces (Methods, Fig. 6g, h). The average recoil velocity and the final maximal displacement of severed lateral interfaces were strongly increased in H/P fold cells that had accumulated lateral F-actin compared to neighboring cells (Fig. 6i, j, Supplementary Movie 9). We conclude that lateral F-actin accumulation in H/P fold cells leads to increased tension along their lateral interfaces, driving pulsatile contractions of cell height and the formation of the H/P fold.

**3D vertex model simulations recapitulate fold formation**. We then asked whether the measured changes in lateral and basal cell edge tension could generate the observed morphological changes and could be sufficient to account for the formation of the H/H and H/P fold. To address this question, we used a 3D vertex model of epithelial mechanics (Fig. 7a[32]), which expands on previous two-dimensional (2D) vertex models[26,33] by describing the apical and basal surfaces of the epithelium in 3D. We consider epithelial mechanics governed by surface and line tensions that are exerted along the cell surfaces and edges. Elastic springs maintain the connection of basal vertices to the ECM (Fig. 7a). Cells maintain their volume while changing shape. To constrain model parameters, we used the aspect ratio of wing imaginal disc cells prior to folding to set the initial aspect ratio of cells in simulations (Supplementary Methods). Furthermore, we used the

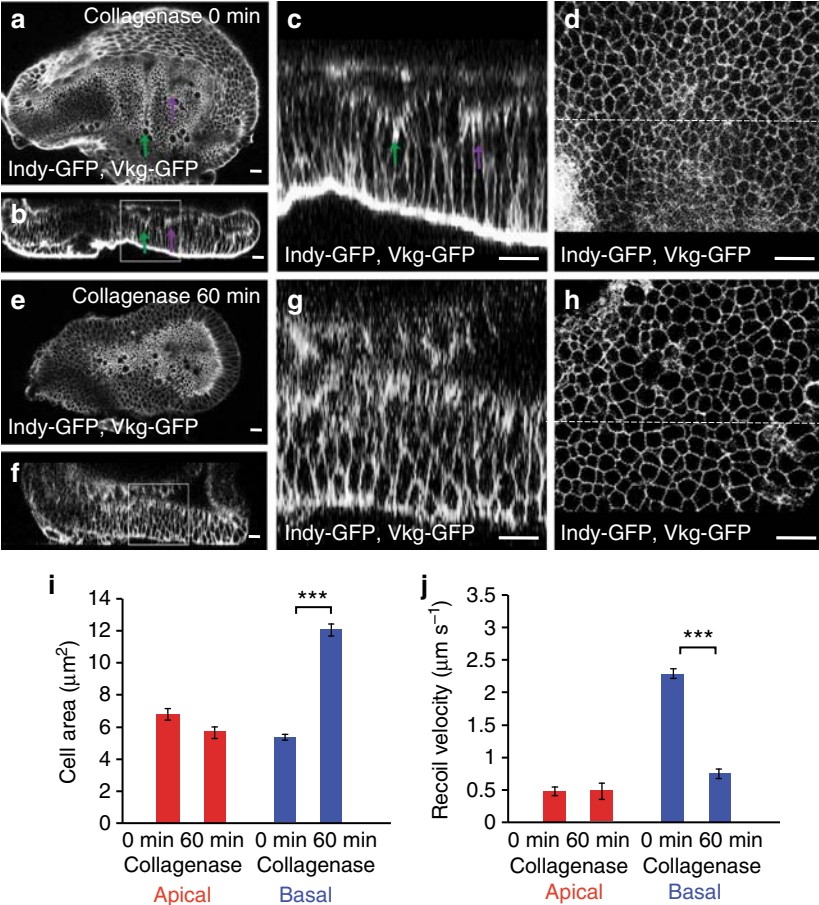

**Fig. 4** Basal tension depends on ECM. **a–h** Apical (**a**, **e**) and cross-sectional (**b**, **c**, **f**, **g**) views of a wing imaginal disc before (**a–d**) and 60 min after (**e–h**) addition of collagenase to the culture medium are shown. Magnifications of the boxed areas are shown in **c** and **g**. **d**, **h** Corresponding basal views. Dotted lines indicate position of cross section. Scale bars are 10 μm. **i** Apical and basal cross-sectional cell area before (0 min) and 60 min after addition of collagenase to the culture medium are shown. Mean and s.e.m. are shown ($n = 365$ (apical, 0 min), 357 (apical, 60 min), 445 (basal, 0 min), and 354 (basal, 60 min) cells of 4 wing imaginal discs) (***$p < 0.001$, Student's $t$-test). **j** Average recoil velocity of the two vertices at the end of an ablated cell edge in the pouch region of 72 h AEL wing imaginal discs before and 60 min after addition of collagenase within 0.25 s after ablation. Recoil velocities are shown for ablations of apical and basal cell edges, as indicated. Mean and s.e.m. are shown ($n = 15$ cuts) (***$p < 0.001$, Student's $t$-test)

experimentally measured ratios of average recoil velocities to constrain ratios of tension parameters (Supplementary Methods). Two free parameters remained, corresponding to the stiffness of basal elastic springs and the ratio of apical and basal edge tension to surface tension. We set normalized versions of these parameters to 1, and found that varying them within a reasonable range did not strongly influence our results (Supplementary Methods, Supplementary Figure 8).

To generate the H/H fold in our simulations, we incrementally decreased the basal surface tension and edge tension of pre-fold cells (basal tension decrease). For the formation of the H/P fold, we incrementally increased the lateral surface tension of pre-fold cells (lateral tension increase) (Fig. 7b, c). In these simulations, we considered a quasistatic folding process, where the system is at any time close to the mechanical equilibrium (see Supplementary Methods); therefore, our model aims at reproducing equilibrium shapes but not the dynamics of folding. We then quantified in our model the same geometric parameters that characterize the morphological changes in the wing imaginal disc. We show them as a function of the mechanical parameters that were changed incrementally, which serves in the quasistatic simulation as an analog of the time axis (compare Fig. 1n–q and Fig. 7c). Remarkably, both basal tension decrease and lateral tension increase led to significant apical invagination of the tissue, with

the shapes of the fold recapitulating the observed experimental shapes (Fig. 7c, d, Supplementary Figure 9a, Supplementary Movie 10). Morphologies of the H/H and H/P folding cells were reproduced by the two sets of simulations, with the H/P fold showing reduced basal expansion and bulging-out compared to the H/H fold (Fig. 7d). Moreover, increased apical tension did not lead to significant folding of the columnar epithelium in our simulations (Supplementary Figure 9b, e–h). We also found that in simulations a larger basal than apical tension (as seen in the wing imaginal disc, Fig. 3m) was contributing to more pronounced folding (Supplementary Fig. 9c–i). Thus, we conclude that a decrease of basal tension alone can explain the formation of the H/H fold, while an increase in lateral tension alone can explain the formation of the H/P fold.

## Discussion

In this work, we have uncovered two new mechanisms of epithelial fold formation. First, a locally defined basal decrease of surface and edge tension, associated with local reduction of ECM density, leads to basal cell expansion and folding. Second, a lateral increase of surface tension at the future fold location, associated with F-actin flows and pulsatile contractions, leads to a local reduction of tissue height and fold formation. It is conceivable that both mechanisms may also operate in combination during epithelial folding.

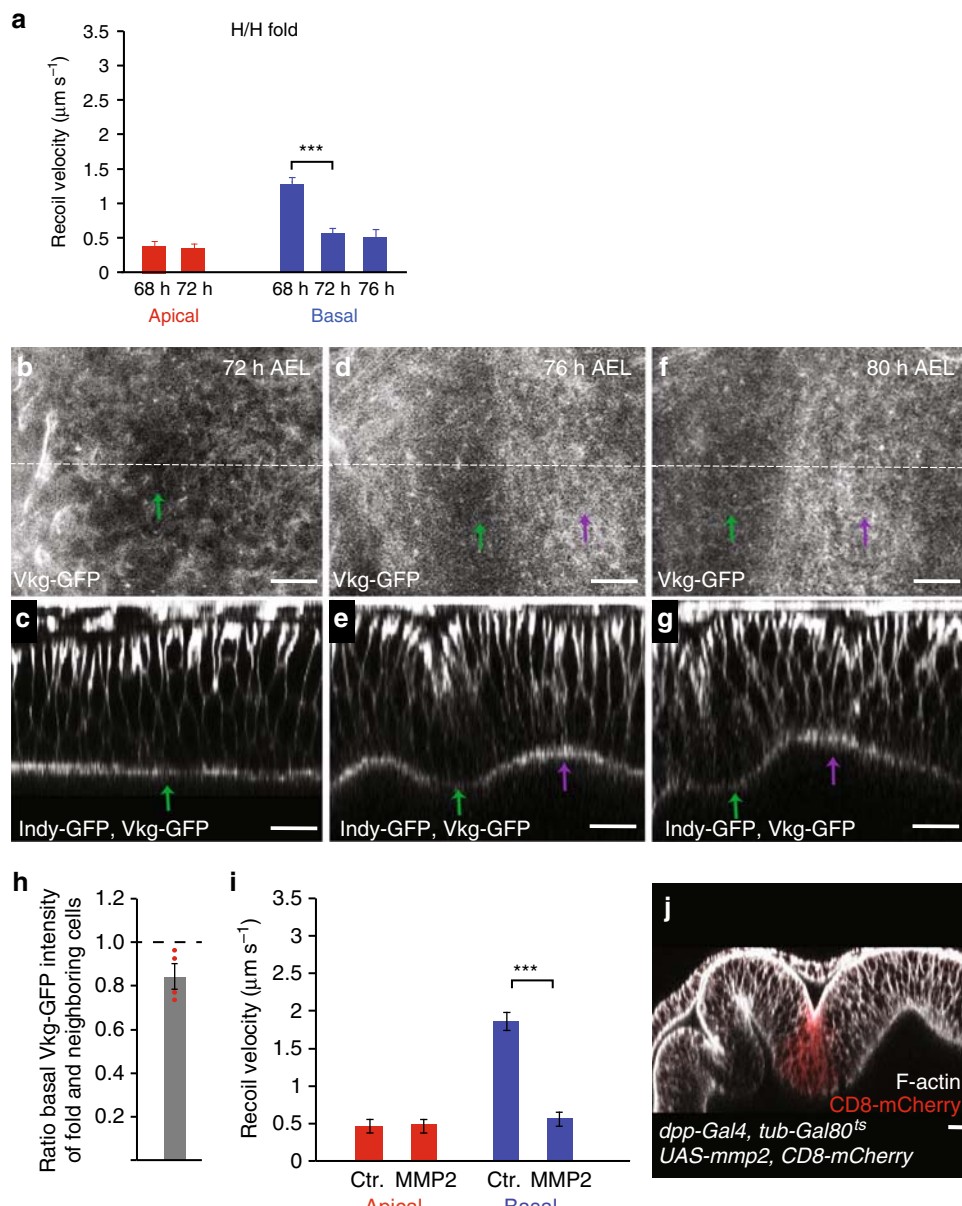

**Fig. 5** Local reduction of ECM and basal tension in H/H fold. **a** Average recoil velocity of the two vertices at the end of an ablated cell edge in the H/H pre-fold region within 0.25 s after ablation for wing imaginal discs of the indicated times AEL. Recoil velocities are shown for ablations of apical and basal cell edges, as indicated. Mean and s.e.m. are shown ($n = 15$ cuts) (***$p < 0.001$, Student's $t$-test). **b–g** Basal (**b**, **d**, **f**) and cross-sectional (**c**, **e**, **g**) views of wing imaginal discs at the indicated stages expressing Vkg-GFP and Indy-GFP are shown. Green and magenta arrows point to the H/H and H/P fold, respectively. Scale bars are 10 μm. **h** Ratio of basal Vkg-GFP pixel intensity for H/H fold cells and neighboring cells of 72 h AEL wing imaginal discs are shown. Mean and s.e.m. are shown ($n = 4$ wing imaginal discs). **i** Average recoil velocity of the two vertices at the end of an ablated cell edge of control cells and cells expressing MMP2 within 0.25 s after ablation. Recoil velocities are shown for ablations of apical and basal cell edges, as indicated. Mean and s.e.m. are shown ($n = 15$ cuts) (***$p < 0.001$, Student's $t$-test). **j** Cross-sectional view of a wing imaginal disc expressing MMP2 in a stripe of cells under control of *dpp-Gal4* labeled by expression of CD8-mCherry (red). F-actin staining is shown in gray. Larvae were incubated for 24 h at 29 °C before dissection to induce MMP2 expression. Scale bar is 10 μm

A simplified picture resulting from our mechanical analysis of how basal tension reduction can induce fold formation is as follows (Fig. 8). Higher basal tension in the cells outside the fold compared to cells inside the fold stretches the basal surface areas of fold cells. Consequently, fold cells widen basally and reduce cell height to maintain cell volume. The new force balance state is characterized by apical indentation and wedge-shaped, shortened cells. How is ECM depletion linked to a decrease in basal cell edge and surface tension? In one scenario, following ECM depletion, the actomyosin network lacks stabilization via binding to integrins, reducing the active tension it can generate with myosin molecular motors. Alternatively, the ECM and cortical actomyosin network, linked together via integrins and other molecules, can be seen as a single composite material under tension[34]. Elastic straining of the ECM, e.g. during tissue growth, could give rise to a passive mechanical tension within the ECM. As the ECM is depleted, the composite material is reorganized and passive ECM stress due to ECM straining could be lost, also contributing to the overall decrease in basal tension in the fold.

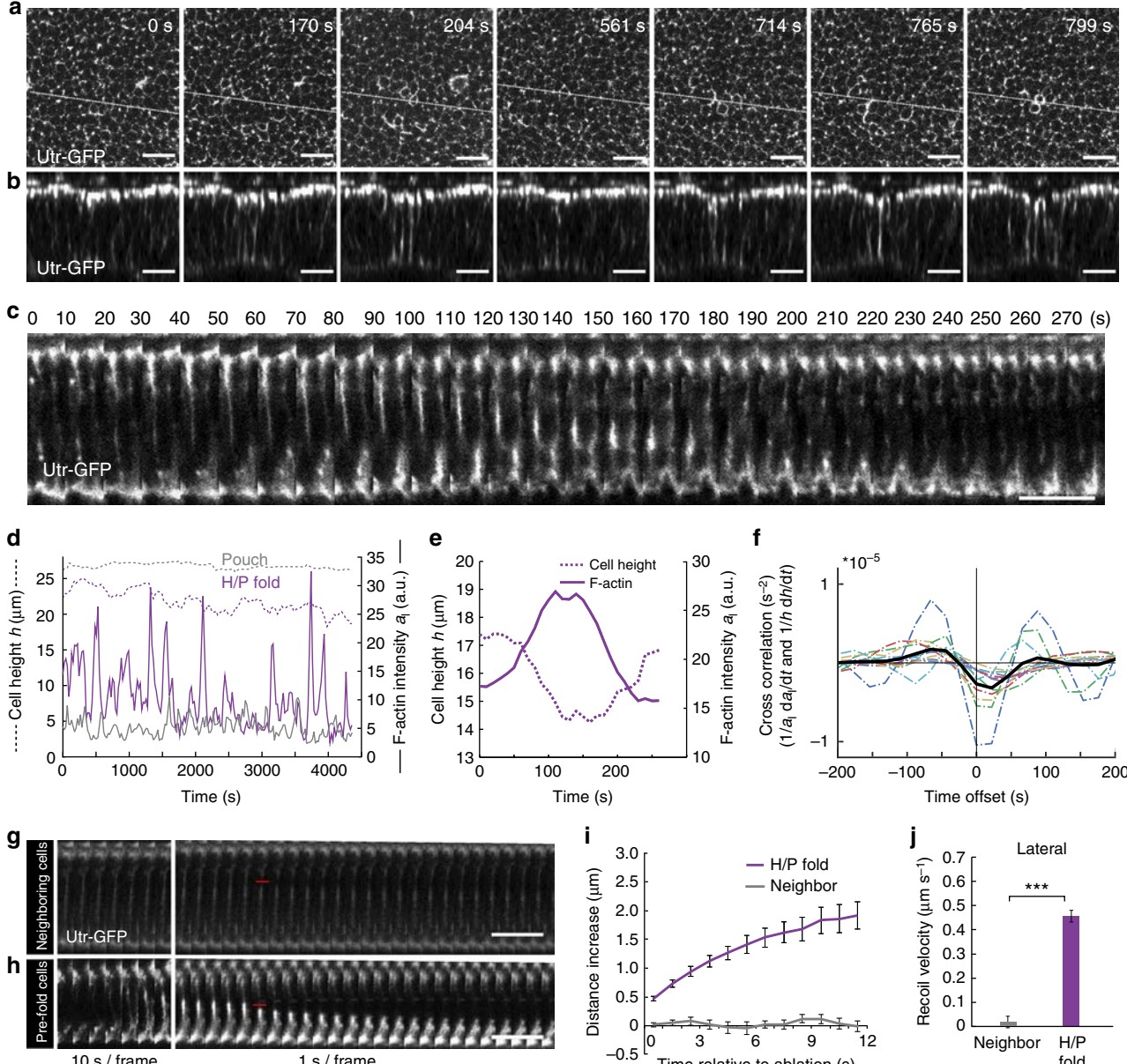

**Fig. 6** Increased F-actin and tension at lateral cell interfaces in H/P fold. **a**, **b** Middle (13 μm below apical surface) *XY* layer (**a**) and cross-sectional images (**b**) of a time-lapse movie of a cultured wing imaginal disc expressing Utr-GFP to label F-actin. The region of the H/P fold is shown. Scale bars are 10 μm. **c** Kymogram of cross sections of Utr-GFP-expressing cells in cultured wing imaginal discs showing the dynamics of F-actin in H/P fold cells. Scale bar is 10 μm. **d** Lateral F-actin intensity $a_l$ (full line) and cell height $h$ (dashed line) for a H/P fold cell (magenta) and a neighboring cell (gray) as a function of time. **e** Close-up view of lateral F-actin intensity $a_l$ (full line) and cell height $h$ (dashed line) for a H/P fold cell as a function of time. **f** Cross correlation function between the relative rate of change of lateral F-actin intensity $(1/a_l)\,da_l/dt$ and rate of relative height change $(1/h)\,dh/dt$ as a function of time offset $\left(\left\langle\left(\frac{1}{a_l}\frac{da_l}{dt}\right)(t)\left(\frac{1}{h}\frac{dh}{dt}\right)(t+\tau)\right\rangle\right.$ as a function of $\tau$). Dotted lines: correlation for twelve individual fold cross sections; black line: average correlation ($n = 12$). The cross correlation is negative for positive time lags and reaches a minimum for a time lag around 22 s. **g**, **h** Kymograms of cross sections of Utr-GFP-expressing neighboring cells (**g**) or H/P fold cells (**h**) before and after ablation of a lateral cell interface. Red lines indicate the time and position of the ablation. Scale bar is 10 μm. **i** Increase of the width of the ablated region along the apical-basal axis upon laser cutting of lateral cell interfaces of H/P fold cells and neighboring cells as a function of time after ablation. Mean and s.e.m. are shown ($n = 15$ cuts). **j** Average recoil velocity within 1 s of ablation of lateral cell interfaces of H/P fold cells and neighboring cells. Mean and s.e.m. are shown ($n = 15$ cuts) (***$p < 0.001$, Student's *t*-test)

Lateral tension increase can also induce fold formation. This can be outlined in a simplified picture (Fig. 8). Increased lateral tension leads to a reduction in cell height. Since basal tension is high, the shortened cells deform the apical surface inwards, while the basal surface resists deformation. As the cells resist volume changes, they widen. Conceivably, increased apical tension in the fold cells favors further basal expansion of the fold cells (see Supplementary Figure. 7a).

Folding requires the transition of cells from a columnar to a wedge-shape where the apical surface is smaller than the basal surface. Previous work has stressed the role of mechanical stresses generated by apical actomyosin networks driving apical constriction during folding[2,9,11]. Our work shows that for the epithelial folds studied here apical constriction is not important. Instead, they rely either on the basal widening of cells due to the decrease of basal tension or alternatively on increased lateral

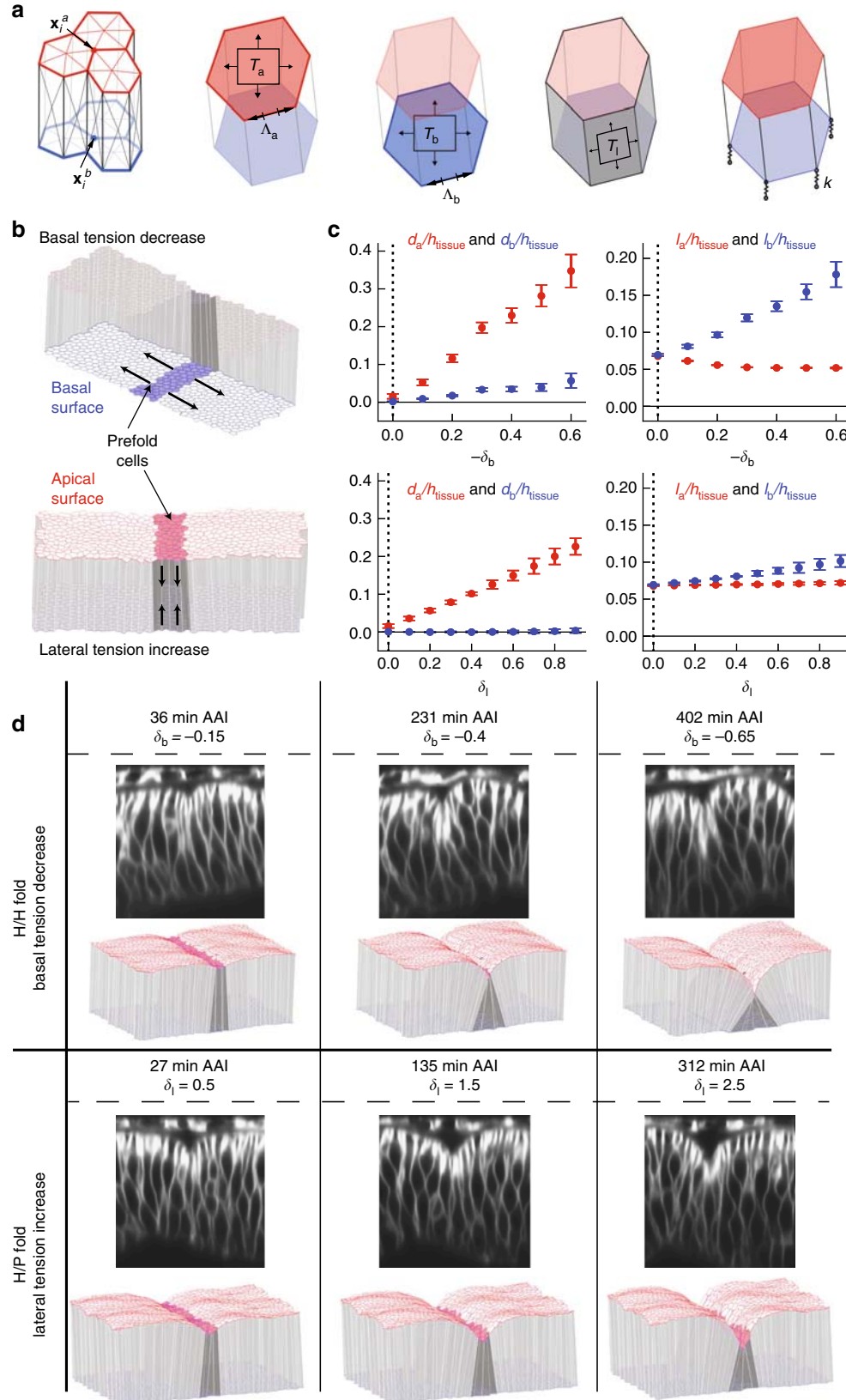

tension. Interestingly, two fundamentally different mechanisms generate similar morphologies of neighboring folds. This implies that the mechanical processes shaping a tissue cannot be deduced from the tissue morphology alone. Cell shortening and an active role for the ECM is also required for the folding of the zebrafish embryonic brain[35]. Basal decrease of tension and lateral increase

of tension may therefore represent two important mechanisms driving the folding of epithelia in different organisms.

## Methods

**Fly stocks and genetics**. The following *Drosophila melanogaster* fly stocks were used: *indy-GFP* (a GFP protein trap in indy (YC0017)[23], *DE-Cad::GFP*[36], *DE-Cad::*

**Fig. 7** 3D vertex model simulations of fold formation. **a** In the 3D vertex model, tissue geometry is represented by a set of apical and basal vertices with positions $\mathbf{x}_i^a$, $\mathbf{x}_i^b$. Cell volume is conserved. In addition, forces acting on vertices arise from apical, basal, and lateral surface tensions ($T_a$, $T_b$, $T_l$) and apical and basal edge tensions at cell–cell contacts ($\Lambda_a$, $\Lambda_b$). Attachment of the basal vertices to the extracellular matrix is represented by elastic springs with spring constant $k$. **b** 3D vertex model representation of the wing imaginal disc epithelium. A packing of identical cells is prepared at mechanical equilibrium, with periodic boundary conditions and mechanical parameters chosen to reproduce the cell aspect ratio in wing imaginal discs. Basal edge and surface tensions are taken four times larger than apical edge and surface tensions. A stripe of pre-fold cells is introduced, with either decreased basal surface and edge tensions $T_b$ and $\Lambda_b$ ("basal tension decrease", upper schematic), or increased lateral surface tension $T_l$ ("lateral tension increase", lower schematic). The tissue configuration is then relaxed to a new state of mechanical equilibrium. **c** Quantification of tissue shape changes in 3D vertex model simulations of fold formation. Geometric parameters (Fig. 1i) as function of the relative decrease of basal edge and surface tension $-\delta_b$ and relative increase in lateral surface tensions $\delta_l$ within pre-fold cells. Mean and s.e.m. are shown ($n = 4$ simulations). Vertical dashed line: initial conditions of simulations prior to fold formation. Basal tension decrease and lateral tension increase lead to folds with a pronounced apical indentation and small basal outward deformations, as observed in H/H and H/P folds (Fig. 1n, o). A more pronounced expansion of basal cell cross-sectional length $l_b$ is observed for the basal tension decrease, similar to the largest basal expansion observed in the H/H fold compared to the H/P fold (Fig. 1p, q). **d** Representative experimental images of H/H (top) and H/P (bottom) folds at successive times, and equilibrium shape of 3D vertex model simulations at increasing magnitude of basal edge and surface tension decrease (top) and lateral tension increase (bottom)

---

mTomato[36], sqh^AX3; sqh-UTR::GFP; sqh-sqh::mCherry[37], viking-GFP[29], Act5C > CD2 > Gal4[38], UAS-CD8-mCherry[39], Cdk1^E1-24 (a temperature-sensitive allele of Cdk1)[25], ap-Gal4[40], UAS-MMP2 (Bloomington Drosophila Stock Center (BDSC) line 58705), UAS-RFP (BDSC line 31417), 30A-Gal4 (BDSC line 37534), doc-Gal4 (BDSC line 46529), dpp-Gal4 (a gift from E. Knust), and tub-Gal80^ts[41].

The genotypes of larvae were as follows:
Figure 1b–g, Supplementary Fig. 1p–q, Supplementary Fig. 2f–i, Supplementary Fig. 10a–c: indy-GFP/Y; DE-Cad::mTomato/DE-Cad::mTomato.

Figure 1j–m, Supplementary Fig. 2a–d, Supplementary Fig. 2n–s, Supplementary Fig. 10d,e: indy-GFP/hsp-flp;; Act5C > CD2 > Gal4, UAS-RFP/+. RFP-marked clones of cells were generated using the FRT-Flp system[42] subjecting 48 h AEL larvae to a 20 min heat-shock at 37 °C. Wing imaginal discs were dissected 24 h after the heat-shock and cultured and imaged in vitro.

Figure 2a: y,w,hsp-flp; DE-Cad::GFP/DE-Cad::GFP; Act5C > CD2 > Gal4, UAS-CD8-mCherry/Act5C > CD2 > Gal4, UAS-CD8-mCherry. Second instar larvae were heat-shocked for 15–20 min at 37 °C and transferred to 25 °C for 48 h before dissection.

Figure 2c–f: DE-Cad::GFP/DE-Cad::GFP. Seventy-two hours AEL instar larvae were raised and dissected at 25 °C. Wing imaginal discs were cultured and immediately imaged at 30 °C for the indicated time periods.

Figure 2g–j: DE-Cad::GFP, Cdk1^E1-24/DE-Cad::GFP, Cdk1^E1-24. Same experimental condition as control.

Figures 3a–c and h, Supplementary Fig. 5a–h, Supplementary Fig. 7b, c, f–h; sqh^AX3; sqh-Utr::GFP/CyO; sqh-sqh::mCherry/sqh-sqh::mCherry.

Figure 3i–l, Supplementary Fig. 3a-c,g-N: indy-GFP/Y; 30A-Gal4, UAS-CD8-mCherry/CyO.

Figures 4a–h and 5b–g: indy-GFP/Y; vkg-GFP/CyO.

Figure 5j, Supplementary Fig. 6j–k: UAS-CD8-mCherry/UAS-mmp2; dpp-Gal4, tub-Gal80^ts/+. Larvae were incubated at 18 °C and transferred to 29 °C for 12 or 24 h before dissection.

Supplementary Fig. 1a–l,n,o: DE-Cad::GFP.

Supplementary Fig. 3d–f,S,T,Y,Z: indy-GFP/Y; doc-Gal4, UAS-CD8-cherry/TM6b.

Supplementary Fig. 3O–R: indy-GFP/Y; UAS-CD8-mCherry/+; dpp-Gal4, tub-Gal80^ts/+. Larvae were incubated at 18 °C and transferred to 29 °C for 12 or 24 h before dissection.

Supplementary Fig. 3U–X: indy-GFP/Y; UAS-CD8-mCherry/UAS-mmp2; dpp-Gal4, tub-Gal80^ts/+. Larvae were incubated at 18 °C and transferred to 29 °C for 12 or 24 h before dissection.

Supplementary Fig. 6a–i: indy-GFP/Y.

**Immunohistochemistry and imaging of fixed samples**. Wing imaginal discs were dissected, fixed, and stained according to standard protocols[43]. Primary antibodies used were rat anti-DE-cadherin (DCAD2, Developmental Studies Hybridoma Bank (DSHB); 1:50) and mouse anti-PSβ-integrin DSHB (1:200). Secondary antibodies, all diluted 1:200 (Molecular Probes) were anti-mouse Alexa 633 and anti-rat CY5. Alexa Fluor 488 phalloidin (Molecular Probes; 1:200) and rhodamine phalloidin (Molecular Probes; 1:200) were used to detect F-actin. For imaging fixed samples, wing imaginal discs were mounted using double-sided tape (Tesa 05338, Beiersdorf, Hamburg, Germany) as spacer between the microscope slide and the coverslip to avoid flattening of the tissue. Images were acquired on a Leica SP5 MP. Image stacks from apical to basal were taken with sections 1 μm apart.

**Time-lapse imaging**. Flies were raised on apple juice plates in cages; eggs were collected at 2 h intervals and incubated at 25 °C. Hatched larvae were fed on standard food until the proper stages. Wing imaginal discs were dissected and cultured in supplemented Grace's medium[44,45]. Grace's medium (Sigma-Aldrich, G9771) was prepared according to the manufacturer's instruction, the pH was adjusted to ~6.7 at room temperature (using 1 M NaOH) and the medium was then filter-sterilized. Grace's medium was supplemented with 5% fetal bovine serum, 1% penicillin-streptomycin, and 1% BIS-TRIS (using a 500 mM stock solution). Ecdysone (Sigma-Aldrich, 20-hydroxyecdysone H5142) was stored in a stock solution of 0.02 mM at −20 °C and added to the medium prior to use to a final concentration of 20 nM. Wing imaginal discs were placed in glass-bottomed Petri dishes (Matek). Imaging was performed using a Leica SP5 MP confocal microscope with a ×40/1.25 numerical aperture oil-immersion objective. For long-term imaging of fold formation, image stacks of 30–40 μm were taken every 3 or 5 min with optimal sectioning (1.3 μm). To observe F-actin dynamics, images stacks of 30–40 μm were taken every 17–22 s with optimal sectioning (1.3 μm). To analyze apical F-actin dynamics, 3–6 apical slices were projected; to analyze basal F-actin dynamics, 2–3 basal slices were projected.

For Fig. 6c wing imaginal discs were mounted with their lateral side facing the microscope objective. This enabled to image the cross-sectional (X–Z) plane of the tissue directly and with high temporal resolution (s); it was only performed when such high temporal resolution was required. To mount wing imaginal discs with their lateral side facing the microscope objective, wing imaginal discs were placed in glass-bottomed Petri dishes (Matek) with their lateral side facing the bottom of the dish under a dissection microscope. The position of the wing imaginal disc was fixed by attaching the lateral edge of the notum part and the trachea to the bottom of the dish using double-sided tape. Images were taken by X–Y scanning of the cell lateral surfaces every 10 s using a Multiphoton Laser Scanning Microscope Zeiss 710 NLO equipped with a C-Apochromat ×40/1.2 W objective.

**Drug treatment**. The Rho kinase inhibitor Y-27632 (Sigma) was resuspended in phosphate-buffered saline (PBS) at 25 mM concentration and was used in culture medium at a final concentration of 1 mM.

Latrunculin A (Abcam) resuspended in dimethylsulfoxide at 1 mM concentration was used in culture medium at a final concentration of 4 μM. Collagenase Type I (Sigma-Aldrich, 1% in PBS) was diluted in culture medium to a final concentration of 0.02%.

**Image processing and analysis**. Acquired images were processed and analyzed with Fiji[46] and the custom-made software Packing Analyzer[47]. Seven to eleven slices were projected by the maximum intensity projection method in Fiji or by PreMosa[48]. Cells were segmented, tracked, and their descendants were traced to establish cell lineages using Packing Analyzer.

**Denoizing and restoration of axial resolution of images**. Due to the sensitivity of the wing imaginal disc to light exposure, all volumetric time-lapses were acquired with reduced laser intensity and a limited number of focal planes. While this prevents phototoxicity, the so-acquired raw images display considerable noise and low axial resolution due to undersampling. To improve signal-to-noise and axial image quality, we applied to the data used for Figs. 1b–g, j–m and 7d, and Supplementary Figure 2a–d, f–i, n, p, r a recently introduced machine-learning-based image restoration approach (Content-Aware Image restoration, CARE)[49]. In order to acquire the necessary 3D training data, we imaged several wing imaginal discs for each of the used markers (Indy-GFP, RFP, and E-cad-Tomato) using two microscope settings: the first as described above using low laser power and axial undersampling (reduced-quality), the second with increased laser power and a fourfold increased number of imaged focal planes (high-quality). For each marker, a residual neural network[49] was subsequently trained to predict high-quality volumes from the reduced-quality input. We finally applied these networks to stacks of 2D images of the raw time-lapse data, resulting in improved image volumes that exhibit considerable less noise and show improved axial resolution (Supplementary Figure 10). The Python-Code for training CARE networks is available at [http://csbdeep.bioimagecomputing.com/doc/].

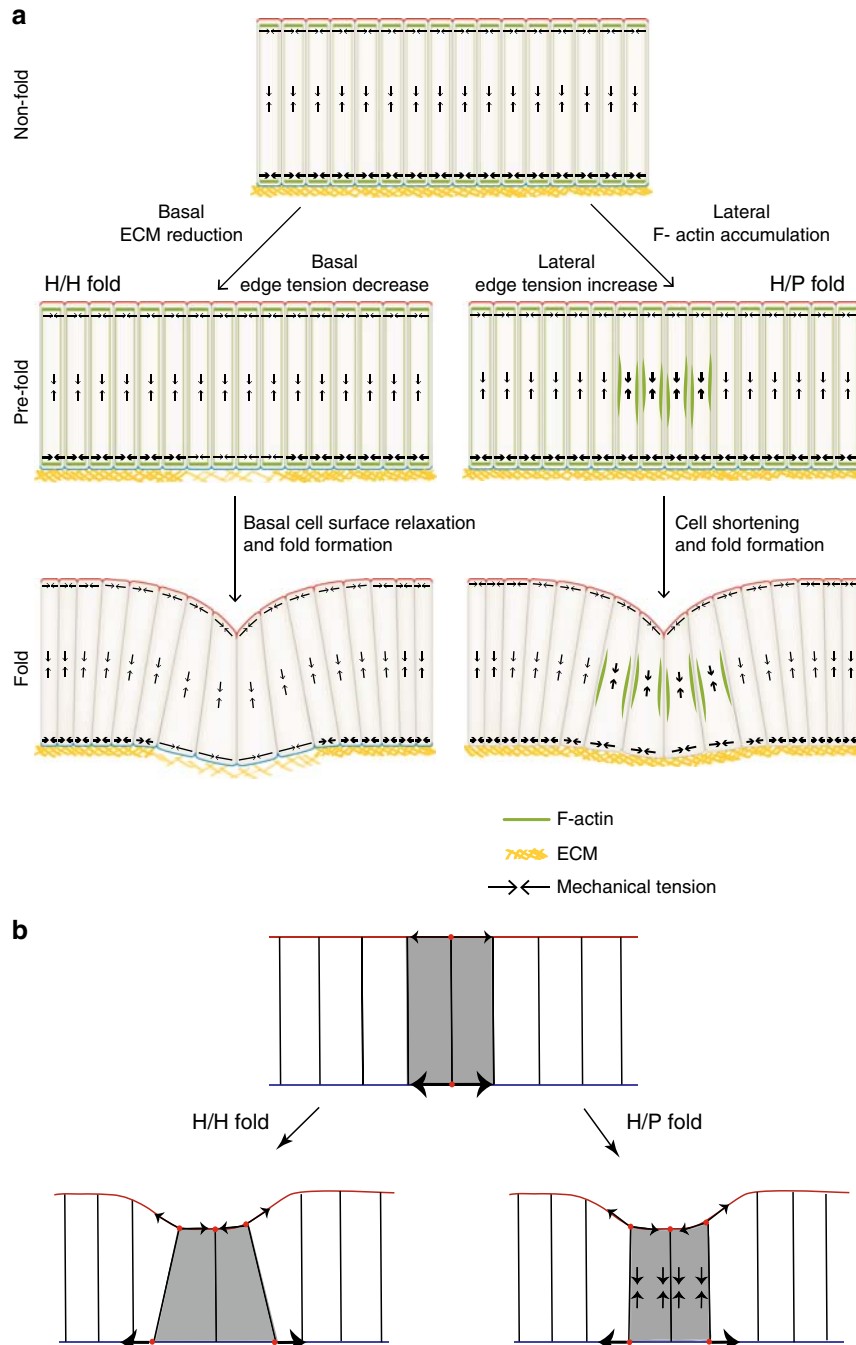

**Fig. 8** Two distinct mechanisms drive H/H and H/P fold formation. **a** Top: scheme of a cross-sectional view of an unfolded epithelium. Note that basal tension is greater than apical tension. Basal tension depends on ECM. The H/H fold and the H/P fold form through two distinct mechanisms. Left: prior to H/H fold formation (pre-fold) a local reduction of ECM leads to a relaxation of basal tension. The decrease of basal tension results in the widening of the basal side of the pre-fold cells; cells adopt a wedge-like shape that drives fold formation. Right: prior and during H/P fold formation, fluctuations of F-actin accumulation at lateral cell interfaces leads to increased lateral tension driving pulsatile cell height contractions. Since apical tension is lower than basal tension, cell shortening leads to apical invagination and fold formation. **b** Simplified picture of mechanism of fold formation. Top: basal tension is greater than apical tension in the unfolded epithelium. Left: in the H/H fold, high basal tension of the neighboring cells stretches the basal surface of the fold cells, in which basal tension is reduced. Cells widen basally and reduce cell height to maintain their volume. Right: in the H/P fold, high lateral tension leads to a reduction in cell height. Since basal tension is high, the shortened cells deform the apical surface inwards, while the basal surface resists deformation

**Apical and basal laser ablation.** For laser ablation experiments, cell edges were visualized by indy-GFP. *30A-Gal4 > UAS-CD8-mCherry* and *Doc-Gal4 > UAS-CD8 mCherry* were used to label the H/H fold and H/P fold, respectively. Wing imaginal discs were mounted in culture medium with their apical side facing the objective for cutting apical cell edges. For cutting basal cell edges, the basal side was facing the objective. An inverted microscope with a ×63/1.2 numerical aperture water-immersion objective equipped with a pulsed, third harmonic solid-state ultraviolet-laser (355 nm, 400 ps, 20 mJ/pulse) was used for ablating single-cell edges. Wing imaginal discs were recorded with a time delay of 0.25 s. The vertex displacement after laser ablation was analyzed with Fiji[46]. The two vertices of the ablated cell edges were manually tracked in the recorded images and the vertex distance increase over time measured. The average recoil velocity $v_0$ was obtained by measuring the vertex distance increase between the time point before ablation and the first image acquired 0.25 s after ablation, and dividing by 0.25 s. The average recoil velocity is taken as a measure of relative mechanical tension on the cell edge before ablation[27] (see Supplementary Methods).

**Laser ablation of lateral cell interfaces**. To ablate lateral cell interfaces, wing imaginal discs were mounted in culture medium with their lateral sides facing to the objective as described above. Images were acquired and laser ablations were performed on a Multiphoton Laser Scanning Microscope Zeiss LSM 710 NLO using a C-Apochromat ×40/1.2 W objective. A lateral cell interface was identified and ablated using a laser beam that created a focal volume with a length of approximately 2 μm and a width of approximately 0.3 μm. The ablation was performed with approximately 60–70 mW of average power (50%) at 800 nm. Utr:: GFP was used to label the lateral cell interfaces. Images were taken by $X–Y$ scanning of the cell lateral surfaces every 10 s before ablation and every 1 s after ablation.

**Quantification of the shape of RFP-marked cells**. Several $Y–Z$ cross sections perpendicular to the folds were generated from acquired movies by Fiji[46]. Apical and basal vertices of RFP-marked cells located in the center of the fold (i.e. two cells on either side of the middle of the fold) were manually tracked over time. The average apical and basal cross-section lengths ($l_a$ and $l_b$) of these cells were then extracted from the tracking using Matlab. The apical and basal indentations ($d_a$ and $d_b$) and the height of cells outside the folds ($h_{tissue}$) were then extracted by tracking the apical and basal outline of the tissue according to the cell membrane marker Indy-GFP.

**Quantification of cell shape**. The apical plane was identified by focusing the image plane on the DE-Cad::mTomato signal of the neighboring cells. The fold plane was identified by focusing the image plane on the DE-Cad::mTomato signal of the cells at the center of the fold. The basal plane was identified by focusing the image plane on the basal surface of fold cells or neighboring cells. Cell meshes in the apical, fold, and basal plane were then segmented and tracked over time using Packing Analyzer[47]. Cell areas were measured using Packing Analyzer[47] or Fiji[46]. Cell areas were measured where cell apical or basal outlines were entirely visible in a single Z-slice, to ensure that the true apical or basal area was measured. Pre-fold cells were identified by tracking cells inside folds back in time.

**Quantification of single-cell volume**. RFP-marked clones consisting of approximately one to three cells were generated using the FRT-Flp system[42]. Wing imaginal discs carrying clones that localized to the fold region were cultured and imaged in vitro. Cell outlines were labeled by Indy-GFP. Z-stacks of 30 slices were acquired from apical to basal to contain the whole cell volume. Clone outlines were manually tracked for each slice from apical to basal according to the clone marker RFP using the plug-in Volume manager of Fiji[46]. The volume of clones was quantified using Volume manager. Single-cell volume was calculated by dividing clone volume by the number of cells per clone. Cell volume was visualized by the plugin 3D Viewer of ImageJ[50].

**Quantification of clone size**. We projected 5–8 apical Z-stacks by maximum intensity projection to obtain the apical cell mesh. The cell number of clones located in the notum or pouch region of the wing imaginal disc was then manually counted.

**Quantification of wing imaginal disc cell number**. The apical cell mesh of cells was obtained by first projecting 5–8 slices of Z-stacks showing DE-Cad::mTomato using the maximum intensity projection tool in Fiji[46]. The first projected movie frame was then segmented using Packing Analyzer[47]. The initial number of cells in this movie frame was calculated by this software. The number of dividing cells in subsequent movie frames was manually counted.

**Measurements of Vkg-GFP levels**. To quantify Vkg-GFP intensities per cell at the basal surface, we segmented the basal side of the wing imaginal disc based on Indy-GFP fluorescence using Packing Analyzer[47]. We then projected 3–5 basal Z-slices of the image stacks by maximum intensity projection to obtain the basal Vkg-GFP intensity images. The Vkg-GFP intensity images were then overlaid with the cell segmentation. Vkg-GFP pixel intensities were then measured in each segmented fold cell and each segmented neighboring cell.

**Measurements of F-actin levels**. To quantify F-actin levels at the lateral interface of single cells, F-actin dynamics was visualized by sqh-UtrophinABD::GFP, and wing imaginal discs were mounted with apical face to the objective (Fig. 6a, b, d, f). Image Z-stacks were taken from apical to basal every 17–22 s. $Y–Z$ cross sections that were generated by Fiji were analyzed. For Fig. 6c, e, wing imaginal discs were mounted with the lateral side facing to the objective. F-actin intensity was measured over time using Fiji by drawing a rectangular region of size 7.3 μm by 14.6 μm that covered the lateral surface of the cell of interest. Cell height was measured over time using Fiji[46] by tracking apical and basal vertices of the cell of interest.

To quantify F-actin levels in the medial apical surface of single cells (Supplementary Fig. 7f-h), wing imaginal discs were mounted with their apical side facing the objective. Image Z-stacks were taken from apical to basal. To quantify F-actin levels in medial basal surface of single cells, wing imaginal discs were

mounted with their basal side facing the objective. Image Z-stacks were taken from basal to apical. In all, 3–5 apical or basal Z-stacks were projected by the maximum intensity projection method. Medial F-actin intensity and cell area were measured over time using Fiji[46] by manually identifying the contour of the cell and extracting the areas and average F-actin intensities.

**Statistical analysis**. A two-sample, unpaired Student's $t$-test was used for statistical analysis.

## Data availability

All the data supporting the findings of this study are available within this paper and its supplementary information.

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

## Acknowledgements

We thank C. Blasse for providing access to PreMosa before publication and S. Shvartsman, K. Röper, T. Xu, Bloomington Drosophila Stock Center, and Vienna Drosophila Resource Center for fly stocks. S.A. and G.S. were supported by the Francis Crick Institute which receives its core funding from Cancer Research UK (FC001317), the UK Medical Research Council (FC001317), and the Wellcome Trust (FC001317). Fl.J. was supported by grant JU 3110/1–1 of the Deutsche Forschungsgemeinschaft. E.W.M acknowledges support by BMBF grant 031L0044 "Sysbio II: Tissue and Organ Formation: A systems microscopy approach".

## Author contributions

L.S. performed all experiments and analyzed data. S.A. performed and analyzed the simulations and contributed to the analysis of experimental data. M.W. performed the image denoising using CARE. N.D. contributed the protocol for cultivating wing imaginal discs. M.W., Fl.J., and E.W.M. contributed image analysis tools. L.S., S.A., S.E., Fr.J., G.S., and C.D. contributed to the design and interpretation of experiments and simulations. L.S., S.A., Fr.J., G.S., and C.D. wrote the manuscript with contributions from all authors.

## Additional information

**Competing interests:** The authors declare no competing interests.

