## [Peer Review File · Nature Communications]

Reviewers' Comments:

Reviewer #1:

Remarks to the Author:

Epithelial folding in diverse animal systems is driven in large part by apical constriction. Here, Sui, Alt, and coauthors present evidence for two alternative mechanisms based on their studies of two regions of the developing *Drosophila* wing disc. Major findings reported in the manuscript include (1) the lack of apical constriction found after measuring aspects of cell shape over time in both the H/H fold and the H/P fold, (2) the demonstration that in the H/H fold, basal tension starts high and decreases over time due to local loss of ECM, and interestingly, that experimental reduction of ECM in a stripe of cells can locally induce a fold, (3) the H/P fold forms differently, based on high tension in lateral membranes and resulting contraction, and (4) that 3D vertex model simulations can produce folds if given the tissue geometries and relative tensions that were revealed in the experiments.

The finding of novel mechanisms of cell shape change would be significant and would justify publication in a high profile journal, in my view. But I have major concerns about large parts of the manuscript, as detailed below. My inclination is to believe that the authors probably have found folds that form by novel mechanisms, but I don't believe that their data yet demonstrate this, for reasons discussed below.

- The main conclusions of this manuscript depend critically on the ability to measure cell shapes. But I cannot see the basal sides of cells in Fig 1. Can they not be seen by this imaging method? The right-most part of Fig 1c' is shown again in Fig 1d with cell shapes manually segmented. I cannot see that the shapes drawn are justified given what can be seen. Lateral membranes are clearly not straight, and which basal portion of each cell corresponds to which apical one is not clear to me at all, given the pseudostratified appearance of the epithelium. Without the ability to convince readers that cell shapes can be measured accurately, I think that very little of this manuscript can be convincing. I would love to see the authors present a much more convincing set of imaging data and tracing of cell shapes. Ideally, the authors would generate single-cell marked clones to do this (or use a marker that gives scattered single-cell patterns. C855a-gal4 might work.)
- The first major finding of the manuscript is that apical constriction does not occur, but a graph of apical area over time is not presented in Fig 1. In Fig S1I, 'apparent' cell apical area does decrease in H/H fold cells. And the model for the H/H fold (Fig 4d) shows the apical sides shrinking down dramatically, to essentially zero area, indicating that either the model does not replicate the cell shapes well, or that it does and apical sides do in fact shrink over time. This mismatch between model and central claims of the manuscript is problematic.
- Terms need to be defined quantitatively, and it may help to have a 3-dimensional illustration of the tissue or at least of one cell at two timepoints in addition to Fig 1e for this. Terms that need to be defined quantitatively include apical constriction, apparent apical area, projected apical area, cross-sectional surface area, cross-sectional length, and in-plane length (which is defined but I don't understand the definition).
- Somewhere it needs to be noted that most of the results are true of tissue taped to a slide, in culture, flooded with ecdysone, and that the authors speculate that the results are likely to apply in vivo too. The single cell marked clones suggested above might help with comparisons of in vivo conditions to culture conditions.
- The cell proliferation data is unconvincing to me. Cell proliferation rates are not shown to equal the rates elsewhere anywhere, and the *cdc2* experiments do not demonstrate the desired and expected effect on proliferation.
- The ECM experiments include a beautiful 'sufficiency' experiment (expression of MMP2 in a stripe), but no 'necessity' experiments, which would seem relatively straightforward and would help build the model the authors propose.
- The manuscript needs a figure showing force vectors mapped onto a diagram of the epithelium, and the effect of depleting ECM, so that readers can make sense of why removing ECM leads to

the epithelial folding shown. Where does the outward force come from? Is the ECM viewed as elastic and constraining outward forces that are also revealed by laser cuts? And do ruptures occur in tensile networks as in the laser cuts when ECM is depleted? Can such ruptures be shown? Do the authors envision that loss of ECM results in lower basal tension initially, or a dissipation of tension over time as the tissue relaxes into a new shape?

- line 194: "indicating a ratcheting mechanisms" Examining dynamics alone cannot demonstrate that a mechanism exists to prevent increase in cell height (for example in the ratchet tool from hardware stores, it is only by dissecting the tool and seeing the inner mechanism, or by attempting to rotate the tool backwards, that the mechanism is revealed, and not by rotating it forward and occasionally stopping).

- Fig 2a-d needs an explanation of how tissues were mounted to result in the brightness shown (presumably a,b have the apical side closer to the coverslip, and c,d are inverted?)

- Fig 2g: why is the measurement done at the pouch, and not the H/P fold?

- Fig 3g needs a mark for where the laser was focused, and at which timepoint.

- Fig 4d has cells in the bottom right figure with apical surfaces at a highly oblique angle, presumably next to midline cells with tiny apical surfaces. Which were measured in the earlier experiments? Midline cells or nearby cells, which might have been stretched by forces at the midline?

Reviewer #2:

Remarks to the Author:

I find this article interesting. It is original, useful for the community; its claims are well supported, its simulations are adapted to the experiments, and it is written in a clear way.

Here are my remarks :

- Provide more quantitative comparison between shape measurements in simulations and experiments.

- add in conclusion a short sentence mentioning that the different mechanisms identified here (basal, lateral) could combine together, or even combine with the formerly identified apical mechanism.

- Fig 1d, even enlarged on the screen, does not show convincingly that the manually drawn lines do correspond to the raw image, maybe because yellow and white are not easy to distinguish. Indicate which raw image it is (is it the 339 min one?) and show separately the manually drawn lines, then a merge with suitable colors. Add indications if needed.

- Fig 1 f-g" : is the time normalisation of trajectories performed manually or with an automatic fit ?

- While Fig 3 shows a sufficient time resolution, in Supp Fig 4 the time resolution is insufficient to capture the actual initial recoil velocity. Measurements are actually displacements (plateau values) rather than velocities. This should be explicitly mentioned in the methods, and the assumption that it is a proxy for tension should be discussed.

- video 9 : clarify the sentence "Note that the time delay changes from an initial 10 sec. to 0 sec."

- video 10 : the sentence "Note that the sequence of minimization steps does not represent a realistic dynamic representation of the presented processes, since dissipative processes are not taken into account in simulations shown here." is important enough to be moved to (or duplicated in) the main text, while at present it is only mentioned in the supp model.

- Supp model :

references [Bielmeier2016] and [press2007numerical] should be provided;
clarify "apical to basal intersection";
explain why lateral tensions are constant while apical and basal ones are not;
after eq 4, clarify whether springs are attached vertically but free to slide horizontally;
Clarify the sentence "Note that the centers of mass of the surfaces are not taken as degrees of freedom, but are taken into account in the calculation of the forces acting on the single vertices."

Reviewer #3:

Remarks to the Author:

General comments

Epithelial folding is an important morphogenetic deformation in developmental processes such as gastrulation and tube formation. It is therefore of interest to understand the mechanisms by which epithelial cells may enact coordinated shape changes to result in tissue folding. In particular, the role of mechanical forces in tissue folding remains poorly understood.

The present work addresses this question using the larval *Drosophila* wing imaginal disc, an excellent model system for studying conserved mechanisms of growth and patterning. The authors investigate the mechanical processes underlying the formation of the central hinge (H/H) fold and hinge-pouch (H/P) fold in this tissue.

Surprisingly, the authors find that the formation of these folds does not occur through the common mechanism of apical constriction, nor through differential cell proliferation. Instead, the authors find two distinct mechanical mechanisms underlying the formation of these two folds. The H/H fold is found to form due to a local reduction of extracellular matrix, which triggers the decrease of basal edge tension in pre-fold cells and in turn drives tissue folding. In the case of the H/P fold, an increase in lateral actin accumulation instead leads to increased tension along lateral interfaces, which leads to ratcheted cell height contractions and in turn drives folding.

The authors conclude their study by illustrating by means of a cell-based computational model that a decrease of basal tension can indeed suffice to explain H/H fold formation, while an increase of lateral tension alone explains H/P fold formation.

I found this study to be well motivated and, for the most part, clearly communicated. In my view the results are novel and of broad interest to scientists working in the areas of tissue morphogenesis and mechanobiology. Please find below specific comments and queries I had regarding the work.

Specific comments

p.4 "During the first 4 hours of folding, the H/H and the H/P folds underwent pronounced apical indentation at similar velocities (Fig. 1f,g)"

It is not quite clear to me that these figure panels demonstrate the stated observations, since time is normalised for each curve such that the apical indentation increases at the same velocity in all folds.

p.5 "We observed an enrichment of F-actin and non-muscle Myosin II along basal cell edges, similar to the previously described actomyosin-rich apical epithelial belt"

To which previous work(s) are the authors referring here?

p.6 "The average initial recoil velocity of ablated basal cell edges was about 3-5 times higher than the recoil velocity of ablated apical cell edges"

The authors do not explicitly refer to this, but the average initial recoil velocity also appears to be increasing over time, according to Fig. 2g. Can the authors comment on what (if anything) the reader should interpret from this increase?

p.7 "Taken together, we conclude that during H/H fold formation the local reduction of ECM triggers a local decrease of basal edge tension driving the relaxation of the basal cell edges and tissue folding"

The authors do not consider the mechanism underlying this reduction of ECM. Are there any likely actor(s) for such patterned ECM reduction, based on what is known in the literature, or is this simply unknown?

p.8 "H/P fold cell height decreased over longer time scales"

By "cell height", are the authors referring here specifically to the measure "h" defined in Fig. 1e?

p.9 "We conclude that increased lateral actin accumulation in H/P fold cells leads to increased tension along their lateral interfaces, driving pulsatile and ratcheted contractions of cell height and the formation of the H/P fold"

I don't completely follow the logic here, specifically how the reduction in cell height leads to fold formation; does this follow e.g. from the fact that cells are 'tethered' basally and their volume is conserved?

p.9 "Cells maintain their volume while changing shape"

The authors assert this without citation. Has it been established in the literature that cell volume is approximately constant throughout fold formation?

p.9 "To generate the H/H fold in our simulations, we incrementally decreased the basal surface tension and edge tensions of pre-fold cells"

Is a slow, progressive increase in these parameter values actually required for fold formation in this model? Or could one also achieve fold formation by 'switching' these parameters to high values over a short timescale? What (if any) constraints there are on the timescale of patterning these mechanical properties, for fold formation to occur?

p.9 "For the formation of the H/P fold, we incrementally increased the lateral surface tension of pre-fold cells", p.32 Fig.3h

How should the reader interpret physically the lower curve in Fig. 3h, which depicts the increase in width of the ablated region along the apical-basal axis upon laser ablation of lateral interfaces of cells neighbouring the H/P fold? Does the fact that this curve is approximately constant suggest that these lateral interfaces are not under tension, and if so, what implications does that have for the lateral surface tension parameter in the computational model?

p.10 "increased apical tension did not lead to significant folding of the columnar epithelium in these simulations"

How should this be interpreted physically, in the light of other tissue contexts where increased apical tension does appear to be the factor driving folding? Which parameters would need to be modulated in the computational model for increased apical tension to successfully lead to significant folding - e.g. would the basal 'tethering' need to be switching off?

p.17 "normalized with respect to the length of the fold at the initiation of apical indentation"

How do the authors actually defined the "initiation of apical indentation", in practice?

p.26 Fig. 1d

It is not obvious to me that the authors' geometric approximation of cell shape, that of polygonal prisms, really reflects the cell shapes observed in the wing disc - at least from the images presented in e.g. Fig. 1, cell shapes seem to be rather more tortorous, and of non-uniform width along the apical-basal axis. Are the authors able to comment on how good an approximation their

prisms are likely to be, particularly with regard to the importance of lateral surface tension identified by the authors for H/P fold formation?

p.26 Fig. 1e, p.27 "average apical and basal cross sectional lengths of cells in the fold"
How do the authors actually choose the end of each fold? E.g. in Fig. 1e, are the immediate neighbours of the cell labelled with I_b considered to be 'in' the fold? This is not quite clear to me. It presumably affects measurements of d_a, d_b, etc.

p.33 "rate of height change $1/h \, dh/dt$ "
Is this "h" the same as the measure "h" defined in Fig. 1e?

Supplemental modeling procedures p.1 "the tissue is allowed to change through topological transitions involving neighbour exchange"
Do the authors actually observe any neighbour exchanges during fold formation, in particular in the fold?

Supplemental modeling procedures p.1 "we assume that apical and basal surfaces have a constant tension if their area is larger than a preset threshold area, and have a linearly elastic behaviour below this threshold"

What is the justification for assuming this specific functional form, rather than e.g. simply assuming a tension that is constant, or that is linear in area?

Possible typos

p.8 "a ratcheting mechanisms similar to" -> "a ratcheting mechanism similar to"
p.10 "with H/P fold showing reduced" -> "with the H/P fold showing reduced"
p.18 "at the lateral interface of single cell" -> "at the lateral interface of single cells"
p.18, p.19 "To quantify F-actin levels in medial" -> "To quantify F-actin levels in the medial"
p.27 "apical nd basal deformations" -> "apical and basal indentations"
Supplemental modeling procedures p.1 "along junction between cells" -> "along junctions between cells"

Reviewer #4:

Remarks to the Author:

The paper by Sui et al. examines the basis for two epithelial folds in the cultured wing discs. Analysis of the H/H fold showed that the apical surface was not reduced. Instead, an increase in the basal surface was responsible for the formation of the fold. A local decrease in the ECM by collagenase or expression of MMP2 was able to mimic this effect. Interestingly, the H/P fold did not seem to alter its basal ECM. Instead, a lateral accumulation of actin followed by ratcheted shortening of the lateral edges seems to be responsible for generating this fold. Vertex models confirm that each of these modes may be sufficient to induce folding.

In general, the paper is important because it presents two alternative modes to epithelial folding, that are distinctly different from the established model of apical constriction which has been widely explored for gastrulation. Furthermore, the capacity of modification of the ECM to alter cell shape offers another tier of regulation for tissue morphogenesis.

The paper is comprehensive, employing detailed measurements, manipulation by laser induced recoil and ECM modification, and is backed by a computational model. It will provide an important contribution to the growing knowledge on the mechanisms driving tissue morphogenesis. I recommend its publication in the present form.

Response to Reviewers

We thank the reviewers for their thoughtful and constructive comments that have helped us to improve our manuscript. Below we detail our response to each of the reviewer's comments.

Reviewer #1 (Remarks to the Author):

Epithelial folding in diverse animal systems is driven in large part by apical constriction. Here, Sui, Alt, and coauthors present evidence for two alternative mechanisms based on their studies of two regions of the developing *Drosophila* wing disc. Major findings reported in the manuscript include (1) the lack of apical constriction found after measuring aspects of cell shape over time in both the H/H fold and the H/P fold, (2) the demonstration that in the H/H fold, basal tension starts high and decreases over time due to local loss of ECM, and interestingly, that experimental reduction of ECM in a stripe of cells can locally induce a fold, (3) the H/P fold forms differently, based on high tension in lateral membranes and resulting contraction, and (4) that 3D vertex model simulations can produce folds if given the tissue geometries and relative tensions that were revealed in the experiments.

The finding of novel mechanisms of cell shape change would be significant and would justify publication in a high profile journal, in my view. But I have major concerns about large parts of the manuscript, as detailed below. My inclination is to believe that the authors probably have found folds that form by novel mechanisms, but I don't believe that their data yet demonstrate this, for reasons discussed below.

- The main conclusions of this manuscript depend critically on the ability to measure cell shapes. But I cannot see the basal sides of cells in Fig 1. Can they not be seen by this imaging method? The right-most part of Fig 1c' is shown again in Fig 1d with cell shapes manually segmented. I cannot see that the shapes drawn are justified given what can be seen. Lateral membranes are clearly not straight, and which basal portion of each cell corresponds to which apical one is not clear to me at all, given the pseudostratified appearance of the epithelium. Without the ability to convince readers that cell shapes can be measured accurately, I think that very little of this manuscript can be convincing. I would love to see the authors present a much more convincing set of imaging data and tracing of cell shapes. Ideally, the authors would generate single-cell marked clones to do this (or use a marker that gives scattered single-cell patterns. C855a-gal4 might work.)

OUR RESPONSE We have generated marked clones of 2-3 cells in the wing disc that allows us to unambiguously identify the shape of cells. (We could not generate single-cell marked clones, because by the time after clone induction the clone marker (RFP) became expressed and detectable, cells had already divided. Nevertheless, we have placed cross sections in a way that we analyse the shape of

single-marked cells (see Fig. 1f,g). Moreover, we have improved image quality by denoising images (see Methods, page 17 of the revised manuscript). As shown in the revised Figure 1f-k, New Supplementary Fig. 2a,b and New Supplementary Video 2, we have traced the shape of single cells in movies of wing disc undergoing folding. Based on these new movies, we have quantified the apical and basal cross-sectional length of cells, as well as apical and basal indentation. Moreover, we have also measured the (true) apical cross sectional area of cells in emerging folds (New Supplementary Fig.2c-g). These data confirm our original conclusion that cells within the Hinge/Hinge and Hinge/Pouch folds do not undergo apical constriction.

We agree with this Reviewer that the analysis of cell shape in marked cell clones is more reliable than the previous segmentation-based analysis. We have therefore replaced the segmentation shown in the original Figure 1 by the new data obtained by analyzing marked clones.

We have also analyzed the expression of C855a-Gal4. While an excellent suggestion, we find it difficult to identify single cells marked by the expression of this Gal4 line (Figure 1 for Reviewer). We therefore resorted to the generation of marked cell clones, as described above.

Figure 1 for Reviewer. Expression of C855a-Gal4 in wing imaginal discs.

(A,B) Apical and (A',B',B'') cross-sectional views of a 76h AEL wing disc expressing CD8-cherry under control of C855a-Gal4 (red). Indy-GFP (green) shows the outlines of all cells in the wing disc. Scale bars are 10 μ m.

- The first major finding of the manuscript is that apical constriction does not occur, but a graph of apical area over time is not presented in Fig 1. In Fig S11, 'apparent' cell apical area does decrease in

H/H fold cells. And the model for the H/H fold (Fig 4d) shows the apical sides shrinking down dramatically, to essentially zero area, indicating that either the model does not replicate the cell shapes well, or that it does and apical sides do in fact shrink over time. This mismatch between model and central claims of the manuscript is problematic.

OUR RESPONSE: We have now measured the ‘true’ apical cross-sectional area of cells in emerging folds (New Supplementary Figure 2c-g). We have been able to do so by using an ‘apical’ marker (E-cadherin-GFP) and by recording images at the apical-basal tissue level where E-cadherin-GFP is visible in x-y sections at the center of the fold (see New Supplementary Figure 2c). Quantification of apical cross-sectional area in H/H and H/P folds shows that it does not shrink over time.

The apical sides of cells as seen in the model for the H/H fold seem small (Fig. 4d), but this is due to the tilted view. We have now added a precise quantification of apical and basal cell area in simulations (see new Supplementary Figure 11a). Our results show that the apical surface area slightly decreases for simulations of basal tension decrease (~15%), and is nearly constant for simulation of lateral tension increase, in agreement with experimental measurements (new Supplementary Figure 2g). The slight discrepancy in cell apical area measured in H/H fold cells (new Supplementary Figure 2f) and in the model for basal tension decrease (new Supplementary Figure 11a) indicates that the model cannot completely capture the folding process. Nevertheless, it is clear that a local reduction of basal tension and a local increase in lateral tension in the simulation captures the main morphological features of Hinge/Hinge and Hinge/Pouch folds (i.e. basal cell widening, apical indentation, no drastic apical shrinkage of cells).

Note that we also realized that the measurement of ‘apparent apical cell area’ was confusing and we have replaced it with the measurements of apical cross-sectional area (see above).

- Terms need to be defined quantitatively, and it may help to have a 3-dimensional illustration of the tissue or at least of one cell at two timepoints in addition to Fig 1e for this. Terms that need to be defined quantitatively include apical constriction, apparent apical area, projected apical area, cross-sectional surface area, cross-sectional length, and in-plane length (which is defined but I don't understand the definition).

OUR RESPONSE: We now show in the revised Figure 1d,e illustrations of the tissue before and during folding. Moreover, we defined the geometrical parameters apical and basal indentation, apical and basal cross-sectional length (Fig. 1e) and apical and basal cross-sectional area (Supplementary Fig. 2e). We no longer use the parameter apparent apical area, because it is confusing and apical cross-sectional area better characterizes the absence of apical constriction.

- Somewhere it needs to be noted that most of the results are true of tissue taped to a slide, in culture, flooded with ecdysone, and that the authors speculate that the results are likely to apply in vivo too.

The single cell marked clones suggested above might help with comparisons of in vivo conditions to culture conditions.

OUR RESPONSE: We mention on page 5 of the manuscript that cell shape is measured in cultured wing discs (“we developed a protocol for live imaging of wing imaginal discs in culture (Methods)”). We also compare the shape of folds in cultured and in in vivo grown wing discs and find that the folds have no visible difference in their shapes (page 5: “formed H/H and H/P folds with no visible difference in shape from the hinge folds of fixed wing discs (Supplementary Data Fig. 1h-i’, Supplementary Video 1)”). We have stated in the Methods section that the analysis of mechanical tension and collagen IV was performed in cultured wing discs.

- The cell proliferation data is unconvincing to me. Cell proliferation rates are not shown to equal the rates elsewhere anywhere, and the *cdc2* experiments do not demonstrate the desired and expected effect on proliferation.

OUR RESPONSE: We use the number of cells per clone as a proxy of relative cell proliferation rate. All clones were generated at a comparable developmental stage. We have revised our analysis and have now plotted the ratio of average number of cells in clones generated in the pouch and average number of cells in clones generated in the notum (revised Supplementary Figure 3b). These data confirm that, at the assayed time window of development, cell proliferation rates in the notum and pouch are similar.

Supplementary Movie 3 shows that upon shift of the wing disc to the restrictive temperature, *cdc2-ts* mutant wing discs no longer undergo cell proliferation (no dividing cells are discernible, in contrast to the control). Yet, fold formation proceeds with similar timing as compared to the control. We therefore maintain our conclusion that cell proliferation is not required for folding.

- The ECM experiments include a beautiful ‘sufficiency’ experiment (expression of MMP2 in a stripe), but no ‘necessity’ experiments, which would seem relatively straightforward and would help build the model the authors propose.

OUR RESPONSE: We use the expression of MMP2 as a *tool* to test whether a reduction of ECM is sufficient to drive fold formation. MMP2 is not expressed (as visualized using a *mmp2-GAL4* line) in fold cells of the wing disc (see Figure 2 for Reviewer). Thus, MMP2 is unlikely to be necessary for fold formation.

Figure 2 for Reviewer: Expression of *mmp2-Gal4*

Apical views of wing discs at the indicated developmental time after egg laying expressing GFP under control of *mmp2-Gal4* (green). F-actin (red) shows the outlines of the emerging folds. Scale bars are 10 μm .

To address whether the reduction of ECM is necessary for H/H fold formation, we had either to prevent ECM degradation (if this is what is happening in the wild-type, as opposed to reduced production of new ECM) or to induce ECM deposition specifically beneath the H/H fold cells. Both experiments are difficult to do. Since we do not know whether (and if yes, how) ECM is locally degraded, we cannot experimentally interfere with this process. ECM deposition is also difficult to alter, because ECM is a multi-protein complex. Moreover, major constituents of the ECM beneath the wing disc cells, including collagen IV, are not produced by the wing disc itself, but rather by circulating hemocytes ('blood cells'). Unfortunately, we cannot experimentally control where hemocytes deposit collagen IV. Thus, while we agree with the referee that a "necessity" experiment would be interesting, we think it is currently technically not feasible.

- The manuscript needs a figure showing force vectors mapped onto a diagram of the epithelium, and the effect of depleting ECM, so that readers can make sense of why removing ECM leads to the epithelial folding shown. Where does the outward force come from? Is the ECM viewed as elastic and constraining outward forces that are also revealed by laser cuts? And do ruptures occur in tensile networks as in the laser cuts when ECM is depleted? Can such ruptures be shown? Do the authors

envision that loss of ECM results in lower basal tension initially, or a dissipation of tension over time as the tissue relaxes into a new shape?

OUR RESPONSE: We have added a new schematic (Supplementary Fig. 12b) and a paragraph in the discussion (page 12) to clarify the mechanisms of fold formation that we propose. Briefly, basal tension reduction leads to an imbalance of forces on the basal side of the fold, which drives a basal area expansion. Consequently, fold cells widen basally and reduce their height to maintain cell volume. Lateral tension increase leads to a reduction of tissue height in the fold. This drives an apical invagination due to the larger resistance to deformation of the basal side.

In our simulations, we only introduce changes of basal and lateral surface tensions to reproduce fold formation, in line with the result of laser ablation experiments. In our simulations, we introduce a possible role for the ECM as providing a passive resisting force against indentation. We show in Supplementary Fig. 11 that this resisting force is not required to generate a fold in our simulations. In addition to playing a passive role, it is also possible that the ECM and cortical actomyosin network can be seen as a single composite material under tension, with elastic straining of the ECM contributing to the overall basal tension. We now discuss this possibility on page 12 of the manuscript. We note however that the mechanism we describe would apply, irrespective of the relative contribution of the ECM and basal actomyosin cortex to basal tension generation.

- line 194: "indicating a ratcheting mechanisms" Examining dynamics alone cannot demonstrate that a mechanism exists to prevent increase in cell height (for example in the ratchet tool from hardware stores, it is only by dissecting the tool and seeing the inner mechanism, or by attempting to rotate the tool backwards, that the mechanism is revealed, and not by rotating it forward and occasionally stopping).

OUR RESPONSE: We agree with the reviewer and have deleted the phrase "indicating a ratcheting mechanism".

- Fig 2a-d needs an explanation of how tissues were mounted to result in the brightness shown (presumably a,b have the apical side closer to the coverslip, and c,d are inverted?)

OUR RESPONSE: The reviewer is correct, a,b have the apical side closer to the coverslip, and c,d are inverted. We now mention this in legend to Fig. 2.

- Fig 2g: why is the measurement done at the pouch, and not the H/P fold?

OUR RESPONSE: We first wanted to test whether mechanical tension is similar or differs between the apical and basal surfaces of the wing disc. We therefore performed the measurements inside the pouch region (outside the folds). This analysis showed that mechanical tension is increased at the basal surface compared to the apical surface. This result has important implications for the mechanism by which the two folds form. As stated on page 12, "Simulations suggest that both

mechanisms lead to significant fold formation only when the basal tension is larger than the apical tension (Supplementary Fig. 9), as seen in the wing disc (Fig. 2g).”

The measurement of basal tension in the H/P fold is shown on Supplementary Fig. 9a. Basal tension in the H/P fold cells is similar to the basal tension of cells outside this fold.

- Fig 3g needs a mark for where the laser was focused, and at which timepoint.

OUR RESPONSE: We have added an asterisk in Fig. 3g to mark the site and time point of laser ablation.

- Fig 4d has cells in the bottom right figure with apical surfaces at a highly oblique angle, presumably next to midline cells with tiny apical surfaces. Which were measured in the earlier experiments?

Midline cells or nearby cells, which might have been stretched by forces at the midline?

OUR RESPONSE: We measured in both experiment (Fig. 1) and simulations (Fig. 4) the shape of the four cells located at the bottom of the fold (i.e. two cell on the ‘right’ and the ‘left’ of the folding midline).

--

Reviewer #2 (Remarks to the Author):

I find this article interesting. It is original, useful for the community; its claims are well supported, its simulations are adapted to the experiments, and it is written in a clear way.

Here are my remarks :

- Provide more quantitative comparison between shape measurements in simulations and experiments.

OUR RESPONSE: We have provided in the manuscript quantitative comparison of the fold apical indentation, basal indentation, apical and basal cross-sectional lengths (Fig 1h-k and Fig. 4c).

Following the request of the referee, we have now added comparisons of the apical and basal areas between experiment and simulations (Fig. S2f-h, Supplementary Fig. 11a).

We note that further comparison of the full time-evolution in the shape changes observed in experiments would involve including an explicit dynamic description of shape changes in our simulations. This would require making additional assumptions on the dissipative mechanisms at play during fold formation. In order to focus on the main mechanisms and avoid having too many unknown parameters, we restricted ourselves here to simulations of quasi-static shapes, which capture the main aspect of the mechanics of fold formation but not its detailed dynamics.

- add in conclusion a short sentence mentioning that the different mechanisms identified here (basal, lateral) could combine together, or even combine with the formerly identified apical mechanism.

OUR RESPONSE: We have added a sentence on page 12 stating that the different mechanisms could also operate in combination during epithelial folding.

- Fig 1d, even enlarged on the screen, does not show convincingly that the manually drawn lines do correspond to the raw image, maybe because yellow and white are not easy to distinguish. Indicate which raw image it is (is it the 339 min one?) and show separately the manually drawn lines, then a merge with suitable colors. Add indications if needed.

OUR RESPONSE: In response to this comment and the comments of Reviewer 1 we have replaced the original segmentation-based analysis of cell shape (original Figure 1) by analyzing the shape of cells in RFP-marked clones of cells. In the new approach (revised Fig. 1f-k), single cells are marked by RFP and thus their outlines can be unambiguously tracked.

- Fig 1 f-g" : is the time normalisation of trajectories performed manually or with an automatic fit ?

OUR RESPONSE: As described above, we have replaced the original segmentation-based analysis of cell shape (original Figure 1) by analyzing the shape of cells in RFP-marked clones of cells. In the new analysis, we have abandoned the time normalization of trajectories and show the morphological parameters as a function of time (in minutes). This allows to better compare the kinetics of fold formation and cell shape change between Hinge/Hinge and Hinge/Pouch folds (see also the first specific comment by Reviewer 3).

- While Fig 3 shows a sufficient time resolution, in Supp Fig 4 the time resolution is insufficient to capture the actual initial recoil velocity. Measurements are actually displacements (plateau values) rather than velocities. This should be explicitly mentioned in the methods, and the assumption that it is a proxy for tension should be discussed.

OUR RESPONSE: For laser cuts of apical and basal cell edges, we have recorded images every 0.25 sec (see Methods). This time resolution is similar to the time resolution that has been previously used to record tissue relaxation upon laser cuts (e.g. Rudolf et al. (2015) Development 142, 3845-3858; Kasza et al. (2014) PNAS 111, 11732-7). We calculate the initial recoil velocity based on the vertex distance increase between the average vertex distance before ablation and the vertex distance measured in the first image acquired 0.25 sec. after ablations (see Methods, page 19). We now mention in the main text on page 7 and in the Methods section that the initial recoil velocity is a proxy for relative mechanical tension on the cell edges before ablation (Mayer et al. (2010) Nature, 467, 617-621).

- video 9 : clarify the sentence "Note that the time delay changes from an initial 10 sec. to 0 sec."

OUR RESPONSE: Sorry for the typo. We corrected this sentence: "Note that the time delay changes from an initial 10 sec. to 1 sec."

- video 10 : the sentence "Note that the sequence of minimization steps does not represent a realistic dynamic representation of the presented processes, since dissipative processes are not taken into account in simulations shown here." is important enough to be moved to (or duplicated in) the main text, while at present it is only mentioned in the supp model.

OUR RESPONSE: We agree with the referee, and have added the following sentence on page 11 of the main text: "In these simulations, we considered a quasistatic folding process, where the system is at any time close to the mechanical equilibrium (see Supplemental modeling procedures); therefore, our model aims at reproducing equilibrium shapes but not the dynamics of folding."

- Supp model :

references [Bielmeier2016] and [press2007numerical] should be provided;

We thank the referee for pointing this out, these references have been added to the Supplemental modelling procedures.

clarify "apical to basal intersection";

We have replaced the previous sentence by "Vertices correspond to the intersection of three or more cells on the tissue apical or basal surfaces".

explain why lateral tensions are constant while apical and basal ones are not;

OUR RESPONSE: We first assumed that the apical and basal tensions were constant but noticed that the homogeneous simulated flat tissue could be unstable to shape undulations with a wave length of the order of a few cells for some choice of parameters. These instabilities appear when the cell shapes are highly columnar, as observed in the wing disc. We therefore chose to introduce an additional elastic contribution to obtain a stable flat tissue shape. We chose to penalize a decrease in cell apical and basal area as a direct way to prevent cell apical and basal area collapse. We have clarified this on page 1 of the supplemental modelling procedures, after Eq. 2.

after eq 4, clarify whether springs are attached vertically but free to slide horizontally;

OUR RESPONSE: We have added the following sentence after Eq. 5 to clarify this point: "With the choice of virtual work made in Eq. 5, we consider elastic bonds that prevent normal deformations away from the ECM but are free to slide tangentially to the plane of the ECM".

Clarify the sentence "Note that the centers of mass of the surfaces are not taken as degrees of freedom, but are taken into account in the calculation of the forces acting on the single vertices."

OUR RESPONSE: We have rephrased this sentence by saying that "when calculating the change in virtual work resulting from the variation of a vertex position in Eq. 6, we include the variations in the position of the centers of mass of surfaces that arise when a vertex of the contour of the surface is displaced". We also give a reference to Bielmeier et al., Curr Bio, 2016, where this point is described in more details.

--

Reviewer #3 (Remarks to the Author):

General comments

Epithelial folding is an important morphogenetic deformation in developmental processes such as gastrulation and tube formation. It is therefore of interest to understand the mechanisms by which epithelial cells may enact coordinated shape changes to result in tissue folding. In particular, the role of mechanical forces in tissue folding remains poorly understood.

The present work addresses this question using the larval *Drosophila* wing imaginal disc, an excellent model system for studying conserved mechanisms of growth and patterning. The authors investigate the mechanical processes underlying the formation of the central hinge (H/H) fold and hinge-pouch (H/P) fold in this tissue.

Surprisingly, the authors find that the formation of these folds does not occur through the common mechanism of apical constriction, nor through differential cell proliferation. Instead, the authors find two distinct mechanical mechanisms underlying the formation of these two folds. The H/H fold is found to form due to a local reduction of extracellular matrix, which triggers the decrease of basal edge tension in pre-fold cells and in turn drives tissue folding. In the case of the H/P fold, an increase in lateral actin accumulation instead leads to increased tension along lateral interfaces, which leads to ratcheted cell height contractions and in turn drives folding.

The authors conclude their study by illustrating by means of a cell-based computational model that a decrease of basal tension can indeed suffice to explain H/H fold formation, while an increase of lateral tension alone explains H/P fold formation.

I found this study to be well motivated and, for the most part, clearly communicated. In my view the results are novel and of broad interest to scientists working in the areas of tissue morphogenesis and

mechanobiology. Please find below specific comments and queries I had regarding the work.

Specific comments

p.4 "During the first 4 hours of folding, the H/H and the H/P folds underwent pronounced apical indentation at similar velocities (Fig. 1f,g)"

It is not quite clear to me that these figure panels demonstrate the stated observations, since time is normalised for each curve such that the apical indentation increases at the same velocity in all folds.

OUR RESPONSE: In response to this comment and the comments by Reviewers 1 and 2, we have replaced the original segmentation-based analysis of cell shape (original Figure 1) by analyzing the shape of cells in RFP-marked clones of cells. In the new analysis, we have abandoned the time normalization of trajectories and show the morphological parameters as a function of time (in minutes) (revised Fig. 1h-k). This allows to better compare the kinetics of fold formation and cell shape change between Hinge/Hinge and Hinge/Pouch folds.

p.5 "We observed an enrichment of F-actin and non-muscle Myosin II along basal cell edges, similar to the previously described actomyosin-rich apical epithelial belt"

To which previous work(s) are the authors referring here?

OUR RESPONSE: We have now added a reference (Farhadifar et al. (2007), *Current Biology* 17, 2095-2104).

p.6 "The average initial recoil velocity of ablated basal cell edges was about 3-5 times higher than the recoil velocity of ablated apical cell edges"

The authors do not explicitly refer to this, but the average initial recoil velocity also appears to be increasing over time, according to Fig. 2g. Can the authors comment on what (if anything) the reader should interpret from this increase?

OUR RESPONSE: We agree with this reviewer that the initial recoil velocity upon ablation of basal cell edges increases over time. We do not know the biological significance (if any) of this increase in tension. The increase in basal tension over time correlates with an increase in overall height of the tissue, but there is no evidence for a functional link. We therefore hesitate to speculate on this in the manuscript.

p.7 "Taken together, we conclude that during H/H fold formation the local reduction of ECM triggers a local decrease of basal edge tension driving the relaxation of the basal cell edges and tissue folding"

The authors do not consider the mechanism underlying this reduction of ECM. Are there any likely

actor(s) for such patterned ECM reduction, based on what is known in the literature, or is this simply unknown?

OUR RESPONSE: The mechanisms underlying the local reduction of ECM are unknown. We have tested the possibility that expression of the matrix metalloproteinase 2 (MMP2), which can degrade ECM components (Supplemental Figure 7d), is specifically expressed in cells of the Hinge/Hinge fold. However, this is not the case (Figure 3 for Reviewer).

Figure 3 for Reviewer: Expression of mmp2-Gal4

Apical views of wing discs at the indicated developmental time after egg laying expressing GFP under control of mmp2-Gal4 (green). F-actin (red) shows the outlines of the emerging folds. Scale bars are 10 μm.

p.8 "H/P fold cell height decreased over longer time scales"

By "cell height", are the authors referring here specifically to the measure "h" defined in Fig. 1e?

OUR RESPONSE: We thank the reviewer for pointing this out. We have changed in Figure 1 the measure "h" to "h_{tissue}" referring to the height of cells neighboring the folds (i.e. the tissue). We use the measure "h" when we refer to the height of cells (e.g. in Figure 3).

p.9 "We conclude that increased lateral actin accumulation in H/P fold cells leads to increased tension along their lateral interfaces, driving pulsatile and ratcheted contractions of cell height and the formation of the H/P fold"

I don't completely follow the logic here, specifically how the reduction in cell height leads to fold

formation; does this follow e.g. from the fact that cells are 'tethered' basally and their volume is conserved?

OUR RESPONSE: We have added a paragraph in the discussion (page 13) and a new schematic (Supplementary Fig. 12) to clarify the mechanism of fold formation by lateral tension increase. Our simulations indicate that a reduction of cell height indeed leads to fold formation as a result of the asymmetry between the apical and basal side of the fold; the fold indents apically either because the basal tension is larger, penalizing basal deformation, or because the tissue is basally tethered to the ECM. This is visible in Fig. S11 (previously Fig. S9): in Fig. S11d, where there is no apico-basal asymmetry, a lateral tension increase does not lead to fold formation. Fold formation by lateral tension increase arises due to ECM attachment (Fig. S11d') or due to a larger basal than apical tension (Fig. S11e).

p.9 "Cells maintain their volume while changing shape"

The authors assert this without citation. Has it been established in the literature that cell volume is approximately constant throughout fold formation?

OUR RESPONSE: We have now measured the volume of cells throughout fold formation and find it to be approximately constant (Supplementary Fig. 2i-l). Interestingly, this is in line with previous work that showed that during tissue folding in the *Drosophila* embryo cell volume remains constant (Gelbart et al., PNAS, 2012, 109, 19298-19393).

p.9 "To generate the H/H fold in our simulations, we incrementally decreased the basal surface tension and edge tensions of pre-fold cells"

Is a slow, progressive increase in these parameter values actually required for fold formation in this model? Or could one also achieve fold formation by 'switching' these parameters to high values over a short timescale? What (if any) constraints there are on the timescale of patterning these mechanical properties, for fold formation to occur?

OUR RESPONSE: In this work, we have performed quasi-static simulations where we calculate mechanically equilibrated shapes following changes in surface tensions (Fig. 4c) and do not take into account dissipative processes. We have clarified this in the main text by adding the following sentence on page 11:

“In these simulations, we considered a quasistatic folding process, where the system is at any time close to the mechanical equilibrium (see Supplemental modeling procedures); therefore, our model aims at reproducing equilibrium shapes but not the dynamics of folding.”

Introducing a full dynamical description of fold formation in our simulation would require to make strong assumptions on the dissipative processes at play in the tissue, which we think are beyond the scope of this manuscript. However, we expect that fold formation could be either achieved by a sudden change in surface tensions and a large viscous resistance to deformation, or by a slow,

progressive change in active forces, as suggested by the referee. Because the recoil observed in laser ablation experiments may arise either from active or viscous stresses in the material (Mayer et al., Nature, 2011), we cannot answer this question at this point. We think it is however a key question for future work and have added a discussion of this point in the Supplemental modelling procedures, in the section “Simulation of epithelial folding”.

p.9 "For the formation of the H/P fold, we incrementally increased the lateral surface tension of pre-fold cells", p.32 Fig.3h

How should the reader interpret physically the lower curve in Fig. 3h, which depicts the increase in width of the ablated region along the apical-basal axis upon laser ablation of lateral interfaces of cells neighbouring the H/P fold? Does the fact that this curve is approximately constant suggest that these lateral interfaces are not under tension, and if so, what implications does that have for the lateral surface tension parameter in the computational model?

OUR RESPONSE: We agree with the referee that a low recoil velocity is observed following ablation of lateral interfaces of cells neighbouring the H/P fold. It seems however unlikely to us that no tension is exerted along the lateral interfaces, as actin and myosin can be observed on these interfaces (Fig. 2a-d'), and the shapes of interfaces is visually consistent with interfaces under tension perturbed by nuclei (see e.g. Fig. 2i'). Possibly, the extent of opening following laser ablation in cells neighbouring the H/P fold cannot be clearly resolved at our microscopy resolution. We have added a discussion of this point in the Supplemental modelling procedures, in the section “Mechanical parameters prior to fold formation”.

p.10 "increased apical tension did not lead to significant folding of the columnar epithelium in these simulations"

How should this be interpreted physically, in the light of other tissue contexts where increased apical tension does appear to be the factor driving folding? Which parameters would need to be modulated in the computational model for increased apical tension to successfully lead to significant folding - e.g. would the basal 'tethering' need to be switching off?

OUR RESPONSE: We have considered in Supplementary Fig. S11 the relative effect of increased apical tension, decreased basal tension, and lateral tension increase. We find that increased apical tension can initiate folding, but the magnitude of indentation is generally less pronounced in the conditions of our simulations, even with basal 'tethering' switched off (left column in Fig. S11). In our simulations, we consider columnar cells with fold regions which are 4 cells wide. It is possible that apical constriction occurring in wider regions, or less columnar cells, induce more significant folding. We have added a sentence at the end of the “Supplemental modelling procedures” to state

this point: "It would be interesting to explore whether changes in forces occurring in larger stripes or in more cuboidal cells lead to different deformations than observed here."

p.17 "normalized with respect to the length of the fold at the initiation of apical indentation"

How do the authors actually defined the "initiation of apical indentation", in practice?

OUR RESPONSE: We define the initiation of apical indentation as the first time point of a time-lapse movie where we see that the apical surface of the fold cells is below the apical surface of the neighbouring cells. We have added a sentence to clarify this definition (page 31, legend to Figure 1b).

p.26 Fig. 1d

It is not obvious to me that the authors' geometric approximation of cell shape, that of polygonal prisms, really reflects the cell shapes observed in the wing disc - at least from the images presented in e.g. Fig. 1, cell shapes seem to be rather more tortuous, and of non-uniform width along the apical-basal axis. Are the authors able to comment on how good an approximation their prisms are likely to be, particularly with regard to the importance of lateral surface tension identified by the authors for H/P fold formation?

OUR RESPONSE: This is an interesting comment of the referee. The wing imaginal disc is a single-cell layered pseudostratified epithelium. All cells contact the apical and basal surface of the tissue and are of uniform height (except for the folds where cells are shorter). The nuclei are located at different apical-basal positions in neighbouring cells. Since the diameter of the cell nucleus exceeds the width of the cell and since nuclei are located at different apical-basal positions, cell shapes are deformed by the nuclei and appear tortuous (see e.g. Fig. 1c'). If the nuclei are homogeneously distributed along the apico-basal axis, we expect that the deformations imposed by nuclei can be treated as a random perturbation which do not change the average mechanical properties of the tissue, and would also not change our conclusions. Adding the deformation imposed by nuclei to our description would be interesting but we think is out of the scope of this manuscript.

p.26 Fig. 1e, p.27 "average apical and basal cross sectional lengths of cells in the fold"

How do the authors actually choose the end of each fold? E.g. in Fig. 1e, are the immediate neighbours of the cell labelled with l_b considered to be 'in' the fold? This is not quite clear to me. It presumably affects measurements of d_a , d_b , etc.

OUR RESPONSE: We have measured the shape of the four middle cells located at the bottom of the fold (as seen on cross sections). This choice does not affect the measurements d_a and d_b . We now state this in the Methods section on page 19.

p.33 "rate of height change $1/h \, dh/dt$ "

Is this "h" the same as the measure "h" defined in Fig. 1e?

OUR RESPONSE: We apologize for the confusion here, in fact it is not the same. As pointed out above, we have changed in Figure 1 the measure “h” to “ h_{tissue} ” referring to the height of cells neighboring the folds (i.e. the tissue). We use the measure “h” when we refer to the height of cells (e.g. in Figure 3).

Supplemental modeling procedures p.1 "the tissue is allowed to change through topological transitions involving neighbour exchange"

Do the authors actually observe any neighbour exchanges during fold formation, in particular in the fold?

OUR RESPONSE: Cell neighbour exchanges have been reported in wing imaginal discs (e.g. Rudolf et al, (2015) Development, 142, 3845-3858), validating the modelling to allow for such topological conditions. We have not specifically analysed the fold cells for neighbour exchanges.

Supplemental modeling procedures p.1 "we assume that apical and basal surfaces have a constant tension if their area is larger than a preset threshold area, and have a linearly elastic behaviour below this threshold"

What is the justification for assuming this specific functional form, rather than e.g. simply assuming a tension that is constant, or that is linear in area?

OUR RESPONSE: We first assumed that the apical and basal tensions were constant, as suggested by the referee, but noticed that for some parameters the homogeneous simulated flat tissue was unstable to shape undulations with a wave length of the order of a few cells. These instabilities can occur in the 3D vertex model when the cell shapes are highly columnar, as observed in the wing disc. We then introduced an additional elastic contribution to obtain a stable flat tissue shape. We chose a non-linear elasticity function that resists cell compression, thus preventing cell apical area collapse, while allowing for cell area expansion.

Possible typos

p.8 "a ratcheting mechanisms similar to" -> "a ratcheting mechanism similar to"

p.10 "with H/P fold showing reduced" -> "with the H/P fold showing reduced"

p.18 "at the lateral interface of single cell" -> "at the lateral interface of single cells"

p.18, p.19 "To quantify F-actin levels in medial" -> "To quantify F-actin levels in the medial"

p.27 "apical nd basal deformations" -> "apical and basal indentations"

OUR RESPONSE: Thank you. We have corrected the typos.

Reviewer #4 (Remarks to the Author):

The paper by Sui et al. examines the basis for two epithelial folds in the cultured wing discs. Analysis of the H/H fold showed that the apical surface was not reduced. Instead, an increase in the basal surface was responsible for the formation of the fold. A local decrease in the ECM by collagenase or expression of MMP2 was able to mimic this effect. Interestingly, the H/P fold did not seem to alter its basal ECM. Instead, a lateral accumulation of actin followed by ratcheted shortening of the lateral edges seems to be responsible for generating this fold. Vertex models confirm that each of these modes may be sufficient to induce folding.

In general, the paper is important because it presents two alternative modes to epithelial folding, that are distinctly different from the established model of apical constriction which has been widely explored for gastrulation. Furthermore, the capacity of modification of the ECM to alter cell shape offers another tier of regulation for tissue morphogenesis.

The paper is comprehensive, employing detailed measurements, manipulation by laser induced recoil and ECM modification, and is backed by a computational model. It will provide an important contribution to the growing knowledge on the mechanisms driving tissue morphogenesis. I recommend its publication in the present form.

OUR RESPONSE: We thank the Reviewer for this positive assessment.

Reviewers' Comments:

Reviewer #1:

Remarks to the Author:

I have read the response to reviewers and the revised manuscript, and I am content with the responses and with the changes made, which I believe have improved the manuscript.

Reviewer #2:

Remarks to the Author:

The authors have well addressed almost all comments. However, one of my central remarks, which questions the whole validity of the tension measurements, has not been adequately addressed. I thus rephrase it below.

In the Supp Fig 5 (formely Supp Fig 4), the time interval between successive images is 0.25 s. This determines the time resolution of the initial recoil velocity measurement. This resolution could be sufficient in some systems and conditions mentioned in the literature. To measure the initial recoil velocity, there must be at least one point (and preferably several points) intermediate between the initial position and the plateau value. In that case, the initial velocity is measured in this intermediate region, using the displacement divided by the time interval (and, in the better case where there are several points, it is measured as the initial slope of the curve). The initial recoil velocity is taken as a proxy for tension measurement, based on the assumption that the dissipation is homogeneous, isotropic and constant. This is a strong assumption, and difficult to check, but is reasonable and commonly accepted in the literature.

However, here, in all 31 curves shown in Supp Fig 5, the resolution is clearly insufficient. What is measured here is the plateau value of displacement, when vertices have reached their equilibrium position. Assuming that it is a proxy for the initial recoil velocity (itself a proxy for tension measurement) is an even stronger assumption. To my knowledge, there is no well accepted justification in the literature. As a referee, I usually accept it when three conditions are simultaneously met: (i) there is no possibility to improve the time resolution, (ii) the epithelium structure is statistically homogeneous and isotropic, and (iii) the results seem plausible.

Here, I question these three points:

- First, I don't know whether the time resolution could have been better, but 0.25 s does not seem too difficult to improve.

- Second, the system structure is very anisotropic because the comparison here is between lateral, basal and apical surfaces: they have a completely different environment, which will affect plateau value displacements.

- Third, because the measurements yield apical:lateral:basal tension ratios of order 1:1:4. However, the newly (and very well) denoised images of lateral surfaces, see Figs 1f,g and Supp Fig 13 a', c', suggest that the lateral surfaces are under much lower tension than the apical ones. I would have been less surprised by ratios 4:1:4, for instance.

Note that the new Supp Fig 4, by showing displacement after 20 sec, does not answer my remark. In fact, I am not sure why in Supp Fig 5 it is useful to plot all points until 20 sec, i.e. 80 points in the plateau region.

Other remarks :

* Line 227

Here and/or in the Supplementary Information, the credit for the vertex model should be given to Honda (Bielmeier should be credited for its 3D version).

Especially since Bielmeier only quotes Farhadifar, and Farhadifar does not quote Honda, so the reader cannot trace the origin of the vertex model.

* Line 420

The denoising procedure is very convincing for the lateral surfaces (Supp Fig 13 a', c').

Has it been applied to 2D sections, 2D projections, whole 3D images ?

Will the code be released ? in 2D and/or in 3D ?

Will it be released with the current training set (to enable the reader to check the current figures) and/or with the possibility to use one's own training set (to enable the reader to apply it to other images) ?

* Line 448

Credit for tension measurement as the initial recoil velocity after ablation should be given to Hutson. I don't see any reason to quote Mayer here.

* Line 524

Explain better how it has been mounted with the lateral side facing the objective.

Explain why it applies to Fig 3d and not to Fig 3c.

Reviewer #3:

Remarks to the Author:

The authors have comprehensively addressed all of my queries and comments.

Although the authors do clearly justify their choice of apical and basal surface tension in their response to one of my original queries (Supplemental modeling procedures p.1), I suggest that it would also be useful for them to include this justification in their supplementary material, so that other readers are not left wondering where this functional form came from.

Reviewer #4:

Remarks to the Author:

The authors have addressed in great detail most of the detailed and insightful comments of referees 1-3. This has improved the paper, which I recommend for publication.

Response to Referees

Reviewer #2

The authors have well addressed almost all comments. However, one of my central remarks, which questions the whole validity of the tension measurements, has not been adequately addressed. I thus rephrase it below.

In the Supp Fig 5 (formerly Supp Fig 4), the time interval between successive images is 0.25 s. This determines the time resolution of the initial recoil velocity measurement. This resolution could be sufficient in some systems and conditions mentioned in the literature. To measure the initial recoil velocity, there must be at least one point (and preferably several points) intermediate between the initial position and the plateau value. In that case, the initial velocity is measured in this intermediate region, using the displacement divided by the time interval (and, in the better case where there are several points, it is measured as the initial slope of the curve). The initial recoil velocity is taken as a proxy for tension measurement, based on the assumption that the dissipation is homogeneous, isotropic and constant. This is a strong assumption, and difficult to check, but is reasonable and commonly accepted in the literature.

However, here, in all 31 curves shown in Supp Fig 5, the resolution is clearly insufficient. What is measured here is the plateau value of displacement, when vertices have reached their equilibrium position. Assuming that it is a proxy for the initial recoil velocity (itself a proxy for tension measurement) is an even stronger assumption. To my knowledge, there is no well accepted justification in the literature. As a referee, I usually accept it when three conditions are simultaneously met: (i) there is no possibility to improve the time resolution, (ii) the epithelium structure is statistically homogeneous and isotropic, and (iii) the results seem plausible.

OUR RESPONSE: In our work, we have measured the displacement at the first time point (250 msec after laser ablation), and have divided by the 250 msec to obtain an initial recoil velocity. We agree with the referee that relaxation nearly saturates at the first time point we measured (250 msec after laser ablation). We therefore, as the referee points out, also measure in effect

the plateau value of displacement, or a value close to the final displacement. We argue below that our measurement of displacement at an early time point is an equally appropriate proxy for tension measurement than initial recoil velocity measurements which have been performed in previous works.

Previous workers have interpreted the relaxation upon ablating single cell junctions using a model where the material is viscoelastic with a Kelvin-Voigt response (Fernandez-Gonzalez et al., 2009; Mayer et al. 2010; Bonnet et al, 2012; Fischer et al., 2014). In this well-accepted model, the recoil velocity depends on the ratio of tension to viscosity, the final displacement on the ratio of tension to elastic modulus, and intermediate displacements at a given time are proportional to the tension, with a proportionality factor which depends on the viscosity, elastic modulus, and time of observation. It is therefore important to see that the initial recoil velocity and the displacement at a given time are equally good measurements of tension, because both are proportional to the tension provided that a material parameter (a viscosity, an elastic modulus or a combination of these two parameters) are constant. As this referee indicates, neither viscosity nor elastic modulus can be directly measured in epithelia, so that both measurements require to make an assumption about the mechanics of the epithelium. In some publications (e.g. Fernandez-Gonzalez et al., 2009; Bonnet et al, 2012; both of which investigated epithelia in *Drosophila*) both the recoil velocity and final displacement were measured. Importantly, both measurements gave rise to the same estimates of ratios of tension. Likewise, Landsberg et al, 2009, has analysed recoil upon single cell junction ablation in *Drosophila* wing disc by measuring recoil velocity, final displacement and radial displacement. All three measurements gave rise to the same estimates of ratios of tension.

Ma et al, 2009 have argued in favour of measuring recoil velocity instead of final displacement. The argument is that above 10s after ablation of a cell junction, changes in the cytoskeleton may start to affect the curve of recoil. We find this a very reasonable argument in general, but if the saturation occurs in 250ms, it is very unlikely that this argument applies. We therefore argue that on sufficiently short time scale, below the timescale of cytoskeletal reorganization, measuring an initial velocity or a displacement at a given time are equally good measurements.

We have revised the manuscript as follows:

-First: We have added a “Discussion of analysis of laser ablation experiments” to the Supplementary Methods discussing in detail how we interpret relaxation in response to ablating cell junctions. In this paragraph, we use a Kelvin-Voigt model to describe the material response following laser ablation, as has been used in previous works (Fernandez-Gonzalez et al., 2009; Mayer et al. 2010; Bonnet et al, 2012; Fischer et al. 2014). We show that within this description, the initial recoil velocity or the displacement measured at a fixed time are equally good estimates of tension variation in the material, as both require that the effective viscosity or effective elastic material are constant to extract tension comparisons from recoil velocities.

-Second: We clarify that our proxy for tension is a displacement at a fixed time point after the laser cut, at an early time point after ablation (250 msec) before changes in the cytoskeleton may start to affect the dynamics of recoil. We explain in the main text of the manuscript on page 7 that this initial displacement (or average velocity of displacement, if one divides by the time span) is a relative measure of mechanical tension.

We would also like to stress that in practice, quantifying an initial recoil velocity is in fact ill-defined, as measurement or intrinsic noise in the vertex displacement makes the calculation of a time derivative of the displacement meaningless. Therefore, all measurements of recoil velocity are in practice measurements of average displacement, divided by the timespan on which the displacement occurs. We have replaced the denomination “initial recoil velocity” by “average recoil velocity” in the manuscript to clarify this point.

Finally, we would like to stress that the three conditions that this referee put forward are met:

1. As detailed below, for technical reasons we cannot improve the time resolution.
2. In most of our comparisons, we compare the results of laser experiments cutting at the same side of cells (either apical, lateral, or basal), where we can safely assume that dissipation is homogeneous and constant. We do make comparisons between cutting apical and basal cell edges, arguing based on the recoil velocity that basal edge tension is higher than apical edge tension. We are aware that the material properties of apical

and basal cell surfaces might differ. We would, however, argue that the ECM present at the basal side is likely a material that increases both viscosity and elasticity of the basal tissue and therefore relaxation velocity and displacement might be *lower* as compared to the apical side. Yet in our laser ablation experiments we find that the recoil velocity is *higher* at the basal side.

3. We think that our results are plausible. First, the higher lateral tension in H/P fold cells is consistent with the increased F-actin levels and reduction in cell height that we observed at the lateral cell interface. Second, the lower basal tension in H/H fold cells is consistent with the relaxation of basal area in these cells. Third, the higher basal tension compared to the apical tension is consistent with the overall curvature of the wing disc: the disc is bent towards its basal side (e.g. McClure and Schubiger, 2005). Fourth, the higher basal tension compared to the apical tension is consistent with the shape of the folds based on our theoretical considerations and simulations using a vertex model. In these simulations, only higher tension at the basal side compared to the apical side results in folds as observed in the wing disc. If apical and basal tension were equally strong, no fold would form; if apical tension were stronger, the fold would form in the opposite direction (Supplementary Figure S9).

Here, I question these three points:

- First, I don't know whether the time resolution could have been better, but 0.25 s does not seem too difficult to improve.

OUR RESPONSE: Unfortunately, it is technically not possible for us to increase the frequency of image acquisition. Using state-of-the-art microscope equipment (a fast spinning disc microscope), the acquisition time for a single image with a sufficient quality to clearly identify cellular vertices (which is required for our analysis) is just below 250 msec. Thus, we cannot go faster than 4 images per second.

- Second, the system structure is very anisotropic because the comparison here is between lateral, basal and apical surfaces: they have a completely different environment, which will

affect plateau value displacements.

OUR RESPONSE: As we argued above, the ECM present at the basal side of cells is likely, if anything, resisting recoil, yet we observe increased recoil velocity there.

- Third, because the measurements yield apical:lateral:basal tension ratios of order 1:1:4. However, the newly (and very well) denoised images of lateral surfaces, see Figs 1f,g and Supp Fig 13 a', c', suggest that the lateral surfaces are under much lower tension than the apical ones. I would have been less surprised by ratios 4:1:4, for instance.

OUR RESPONSE: It seems to us that the referee has taken our data from the measurements of recoil in response to ablating lateral cell edges in the H/P fold (Fig. 3i) to derive apical:lateral:basal tension ratios in the tissue of the order 1:1:4. However, as stated in the manuscript on page 11, recoil in response to ablating lateral edge tension is only strong in cells forming the H/P fold and only during a short period of lateral F-actin flow. Indeed, outside the H/P fold, relaxation in response to ablations of lateral cell edges is very low (see Fig. 6i,j), which is consistent with the expectations of this reviewer (and ours) based on the shape of the lateral cell surfaces.

Note that the new Supp Fig 4, by showing displacement after 20 sec, does not answer my remark. In fact, I am not sure why in Supp Fig 5 it is useful to plot all points until 20 sec, i.e. 80 points in the plateau region.

OUR RESPONSE: We have plotted all points until 5 sec only.

Other remarks :

** Line 227*

Here and/or in the Supplementary Information, the credit for the vertex model should be given to Honda (Bielmeier should be credited for its 3D version).

Especially since Bielmeyer only quotes Farhadifar, and Farhadifar does not quote Honda, so the reader cannot trace the origin of the vertex model.

OUR RESPONSE: We have quoted the paper by Honda (Honda and Eguchi, J. Theor. Biol. (1980) 84, 575-588).

* Line 420

The denoising procedure is very convincing for the lateral surfaces (Supp Fig 13 a', c').

Has it been applied to 2D sections, 2D projections, whole 3D images ?

Will the code be released ? in 2D and/or in 3D ?

Will it be released with the current training set (to enable the reader to check the current figures) and/or with the possibility to use one's own training set (to enable the reader to apply it to other images) ?

OUR RESPONSE: We now explain in the Methods section on page 20 that the denoising procedure has been applied to stacks of 2D images. We also provide on this page a link to the Python-Code for training CARE Networks.

We will release the current training set upon publication of the paper.

* Line 448

Credit for tension measurement as the initial recoil velocity after ablation should be given to Hutson. I don't see any reason to quote Mayer here.

OUR RESPONSE: We have quoted Hutson instead (Ma, ..Hutson, Phys. Biol. 6 (2009) 036004).

* Line 524

Explain better how it has been mounted with the lateral side facing the objective.

Explain why it applies to Fig 3d and not to Fig 3c.

OUR RESPONSE: We have provided a better explanation how we have mounted the wing disc with the lateral side facing the objective on page 18 in the Methods section.

We also explain on this page that this mounting procedure applies to Fig. 3d, because the time resolution shown in this figure is much higher as compared to Fig. 3c. Imaging with high time resolution (seconds) requires to mount the wing disc with the lateral side facing the objective. This enables to image the cross-sectional (xz) plane of the tissue directly. For imaging at low

time resolution, as in Fig. 3c, we use conventional xy imaging to record image stacks. Based on the image stacks, we then construct the cross-sectional view.

References:

- Bonnet et al, 2012, *J.R.Soc.Interface* 9, 2614-2623; doi:10.1098/rsif.2012.0263
Fernandez-Gonzalez et al. 2009, *Dev. Cell* 17, 736-743; doi: 10.1016/j.devcel.2009.09.003
Fischer et al, 2014. *PLoS One*, 9(4), p.e95695; doi: 10.1371/journal.pone.0095695
Landsberg et al., 2009, *Curr. Biol.* 19, 1950-1955; doi: 10.1016/j.cub.2009.10.021
Ma et al. 2009, *Phys. Biol.* 6, 036004; doi:10.1088/1478-3975/6/3/036004
Mayer et al. 2010, *Nature* 467, 617-621; doi:10.1038/nature09376
McClure and Schubiger 2005, *Development* 132, 5033-5042; doi:10.1242/dev.02092

Reviewer #3

The authors have comprehensively addressed all of my queries and comments.

Although the authors do clearly justify their choice of apical and basal surface tension in their response to one of my original queries (Supplemental modeling procedures p.1), I suggest that it would also be useful for them to include this justification in their supplementary material, so that other readers are not left wondering where this functional form came from.

OUR RESPONSE: The Supplemental modelling procedures are now part of the Supplementary Information.